# Looking into User's Long-term Interests through the Lens of Conservative Evidential Learning

**Dingrong Wang**[1], **Krishna Prasad Neupane**[2], **Ervine Zheng**[3], **Qi Yu**[1]*
[1]Rochester Institute of Technology, [2]Amazon, [3]Samsung Research
`{dw7445,kpn3569, mxz5733, qi.yu}@rit.edu`

## Abstract

Reinforcement learning (RL) provides an effective means to capture users' evolving preferences, leading to improved recommendation performance over time. However, existing RL approaches primarily rely on standard exploration strategies, which are less effective for a large item space with sparse reward signals given the limited interactions for most users. Therefore, they may not be able to learn the optimal policy that effectively captures user's evolving preferences and achieves the maximum expected reward over the long term. In this paper, we propose a novel evidential conservative Q-learning framework (ECQL) that learns an effective and conservative recommendation policy by integrating evidence-based uncertainty and conservative learning. ECQL conducts evidence-aware explorations to discover items that are located beyond current observations but reflect users' long-term interests. It offers an uncertainty-aware conservative view on policy evaluation to discourage deviating too much from users' current interests. Two central components of ECQL include a uniquely designed sequential state encoder and a novel conservative evidential-actor-critic (CEAC) module. The former generates the current state of the environment by aggregating historical information and a sliding window that contains the current user interactions as well as newly recommended items from RL exploration that may represent short and long-term interests respectively. The latter performs an evidence-based rating prediction by maximizing the conservative evidential Q-value and leverages an uncertainty-aware ranking score to explore the item space for a more diverse and valuable recommendation. Experiments on multiple real-world dynamic datasets demonstrate the state-of-the-art performance of ECQL and its capability to capture users' long-term interests.

## 1 Introduction

Recommender systems (RS) have been widely used for providing personalized recommendations in diverse fields, such as media, entertainment, and e-commerce by effectively improving user experience (Su & Khoshgoftaar, 2009; Sun et al., 2014; Xie et al., 2018). Increasing efforts have been devoted to capturing users' evolving preferences by shifting the latent user preference over time (Koren, 2009; Charlin et al., 2015; Gultekin & Paisley, 2014). Similarly, sequential recommendation methods (Kang & McAuley, 2018; Tang & Wang, 2018) attempt to incorporate users' dynamic behavior by leveraging previously interacted items. However, existing dynamic recommendation methods primarily focus on maximizing the immediate (*i.e.,* short-term) reward when making recommendations. As a result, they fail to take into account whether these recommended items will lead to long-term returns in the future, which is essential to maintaining a stable user base for the system in the long run.

Several recent works have adapted reinforcement learning (RL) in the RS context (Chen et al., 2019; Zhao et al., 2017). RL has gained huge success in diverse fields, such as robotics (Kober et al., 2013) and games (Silver et al., 2017). The core idea of RL is to learn an optimal policy to maximize the total expected reward in the long run. RL methods consider a recommendation procedure as sequential interactions between users and RL agents to learn the optimal recommendation policies effectively. Although RL approaches show promising results in RS (Chen et al., 2019; Zheng et al., 2018), they

---

*Corresponding author.

primarily rely on standard exploration strategies (*e.g.*, $\epsilon$-greedy), which are less effective for a large item space with sparse reward signals given the limited interactions for most users. Therefore, they may not be able to learn the optimal policy that effectively captures user's evolving preferences and achieves the maximum expected reward over the long term.

Figure 1 further illustrates the limitation of exploration using standard $\epsilon$-greedy. In particular, we compare a RS model trained using $\epsilon$-greedy and one trained using evidential uncertainty, which are denoted as $\epsilon$-greedy and ECQL, respectively. In each step, both models recommend five items, and we identify if their ground-truth ratings are actually positive or not, by comparing with a positive threshold. The count of positive items in each step defines the **Positive Count**. Also, we collect the genre type information of these items across 12 steps. In the end, we have 60 total items of different genres and use them to generate the **Genre Count** plot.

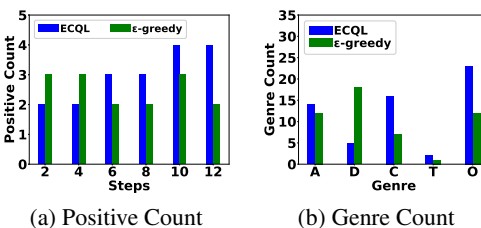

(a) Positive Count          (b) Genre Count

Figure 1: Different recommendation behavior for a representative test user with $\epsilon$-greedy based exploration and our model ECQL with evidential uncertainty (*i.e.,* vacuity).

As can be seen, the existing RL agent primarily focuses on highly-rated items in early steps, as shown in Figure 1a. Most of these items come from a narrower set of genres as shown in Figure 1b, where different genres are denoted as Adventure (A), Drama (D), Comedy (C), Thriller (T), and Others (O). These only represent the user's temporary interest as in the later steps the positive count (See definition in Section 5) of the same genre is decreased. Such a recommendation behavior leads to a lower test return (average rating of recommendation) in the later steps. To further show the advantage of exploration with evidential uncertainty, we define genres (bold) with top-5 average ground-truth ratings as user's long-term interests. Then we capture the last step's recommended items from one test user to see if they match these genres representing user's long-term interests. As Table 1 shows, $\epsilon$-greedy mostly focuses on Drama movies based on the user's short-term interests analyzed from the initial user profile. It only captures one novel genre (*i.e.,* Musical, bold in the table) that matches the user's long-term interest, while ECQL captures four items with more diverse genres that reflect the user's true interests. It clearly indicates that more systematic exploration is essential to discover users' long-term interests to maximize future return.

To address the above key challenges, we conduct novel evidential conservative Q-learning (ECQL) that utilizes a balanced exploitation (with high predicted ratings) and exploration (with evidential uncertainty) strategy for effective recommendations. We formulate an evidential RL framework that augments the reward based RL objective with evidential uncertainty to facilitate the exploration of unknown items. The evidential uncertainty formulation substantially improves exploration and robustness by acquiring diverse behaviors that are indicative

Table 1: Examples of recommended movies

| Model | Important Items (Movies) | Movie Genre |
|---|---|---|
| ECQL | Sound of Music (1965) | **Musical** |
| | Casino (1995) | Drama |
| | Ben-Hur (1959) | **Action** |
| | The Bug's Life (1998) | **Comedy** |
| | Babe (1995) | **Comedy** |
| $\epsilon$-greedy | Pocahontas (1995) | **Musical** |
| | Wizard of Oz (1939) | Drama |
| | Christmas Story (1983) | Drama |
| | Erin Brockovich (2000) | Drama |
| | Restoration (1995) | Drama |

of a user's long-term interest. As shown in Figure 1 (b), ECQL devotes a strong focus on more diverse genres (*i.e.,* 'others' in the figure), and many of these capture the long-term interest from the user as verified by the detailed recommendation list in Table 1. Additionally, we encourage the model to explore items that do not significantly deviate from users' current interests given the sparse interactions. Such gradual exploration is guided by a conservative evidential Q-value that prevents recommending totally irrelevant items, causing user frustrations. We theoretically prove that the conservative evidential Q-value provides an uncertainty-aware adjustment of optimism towards the behavior policy that represents users' current interests. Also, we demonstrate that such Q-value estimation produces a lower bound on the actual value of the target policy, providing a conservative view for safe recommendations. We further show that such a conservative view can be incorporated into a policy learning procedure with theoretically guaranteed policy improvement.

The proposed ECQL seamlessly integrates two major components: a sequential state encoder and a *Conservative Evidential Actor-Critic (CEAC)* module. The former primarily focuses on generating

the current state of the environment by aggregating the previous state, the current items captured by a sliding window, and the future items from the recommendation. It provides an effective means of dynamic state representation for better future recommendations. The CEAC module leverages evidential uncertainty to effectively explore the item space to recommend items that potentially align with the user's long-term interest. It encourages learning the optimal policy by maximizing a novel conservative evidential Q-value to make more diverse recommendations that may reflect a long-term interest while keeping a conservative view that does not deviate too much from current interests. The main contribution of this paper is five-fold:

- a novel recommendation model that integrates reinforcement learning with evidential learning to provide uncertainty-aware diverse recommendations that may reflect users' long-term interests,
- evidential uncertainty guided exploitation to maximize information gain to inform model learning,
- conservative off-policy formulation to avoid over-estimation of policy value that leads to low-quality recommendations due to overfitting to sparse training data from limited user interactions,
- a thorough theoretical analysis to justify the convergence behavior and avoid risky (or overly optimistic) recommendations.
- seamless integration of a sequential encoder, an actor-critic network, and an evidence network to provide an end-to-end integrated training process.

We conduct extensive experiments over four real-world datasets and compare with state-of-the-art baselines to demonstrate the effectiveness of the proposed model.

## 2 RELATED WORK

**Sequential recommendation models.** Recently developed sequential recommendation models utilize historical interactions to capture users' preferences over time through various deep architectures, including CNN (Tang & Wang, 2018), transformer (Kang & McAuley, 2018), and bidirectional encoder (Sun et al., 2019). $S^3$-Rec (Zhou et al., 2020) considers the intrinsic data correlation with mutual information maximization to derive a self-supervised signal to enhance the data representation. CL4SRec (Xie et al., 2022) employs contrastive learning to learn the self-supervised signal from the user behavioral data, which helps to extract more meaningful patterns for user representation. However, sequential models are inadequate to capture the users' long-term preferences. The proposed ECQL model aims to fill this critical gap by performing evidence-guided exploration to maximize the total expected reward over the long run.

**RL-based models.** RL-based RS models aim to learn an effective policy to capture a user's latent preference. The on-policy learning with contextual bandit (Li et al., 2010b) and Markov Decision Process (MDP) (Zheng et al., 2018) exploits by interacting with real customers in an online environment. A collaborative contextual bandit algorithm called CoLin (Wu et al., 2016) utilizes a graph structure in a collaborative manner. On the other hand, off-policy RL utilizes Monte Carlo (MC) and temporal-difference (TD) methods to achieve stable and efficient learning with users' history (Farajtabar et al., 2018). A Q-network with a hierarchical LSTM has been developed to optimize long-term user engagement (Zou et al., 2019). Similarly, model-based RL models user-agent interactions via a generative adversarial network (Bai et al., 2019). Pseudo Dyna-Q further integrates both direct and indirect RL approaches in a single unified framework without requiring real customer interactions (Zou et al., 2020). More recently, SAR leverages an actor-critic network, where the action is generated as an adaptive sequence length to better represent the user's sequential (preference) pattern (Antaris & Rafailidis, 2021). ResAct utilizes a residual actor network to reconstruct a policy that is close to but more efficient than online policy in sequential recommendations (Xue et al., 2022). However, most existing RL based models rely on random exploration strategies, which are less effective at capturing users' long-term preferences, especially from a large item space with sparse user interactions. In contrast, our ECQL utilizes evidence-based uncertainty to systematically explore the item space to maximize the long-term reward.

## 3 PRELIMINARIES

**Recommendation Formulation with RL.** We formulate recommendation tasks in an RL setting, where an RL agent interacts with the environment (*i.e.,* users and items) to recommend the next items to a user. We design this problem as an MDP, which includes a sequence of states, actions, and rewards, denoted as $(\mathcal{S}, \mathcal{A}, \mathcal{P}, \mathcal{R})$. A state $\mathbf{s}_t = \text{SSE}(\cdot|\mathbf{s}_{t-1}, \mathbf{u}_t) \in \mathcal{S}$ is generated by a sequential state encoder that utilizes previous state $\mathbf{s}_{t-1}$ and current user embedding $\mathbf{u}_t$ generated from the

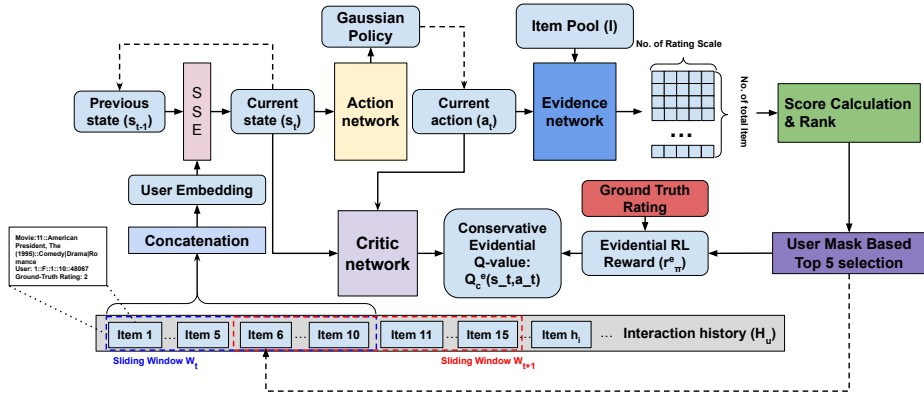

Figure 2: Overview of the ECQL framework

concatenation of $N$ items provided by a sliding window and an RL-agent. An action $\mathbf{a}_t \in \mathcal{A}$ is a vector that represents the user's current and potential preferences that is used to recommend top-$N$ items for a user given the current state $\mathbf{s}_t$. The transition probability $\mathcal{P}(\mathbf{s}_{t+1}|\mathbf{s}_t, \mathbf{a}_t)$ quantifies the probability from state $\mathbf{s}_t$ to $\mathbf{s}_{t+1}$ with an action $\mathbf{a}_t$. The RS environment provides a reward $r_t \in \mathcal{R}$ based on the recommendation quality in the current time step.

**Uncertainty and Evidential Theory.** Subjective Logic (SL) is a probabilistic logic that is built upon probability theory and belief theory (Jsang, 2016). It represents uncertainty by introducing vacuity of evidence in its opinion, which is a multinomial random variable $y$ in a $K$-simplex domain $\mathbb{Y}$. This opinion can be equivalently represented by a $K$-dimensional Dirichlet distribution $\text{Dir}(\boldsymbol{p}|\boldsymbol{\alpha})$ where $\boldsymbol{\alpha}$ is a strength vector over $K$ classes and $\mathbf{p} = (p_1, ..., p_K)^\top$ governs a categorical distribution over $\mathbb{Y}$. The *evidence* $\mathbf{e} = (e_1, \cdots, e_K)^\top$ measures the number of supportive observations from data for each class. It has a fixed relationship $e_k = \alpha_k - 1$ with the Dirichlet strength $\boldsymbol{\alpha}$ given a non-informative prior. Let $e_k$ be the evidence for a class $k$. SL measures different types of second-order uncertainty through evidence, including vacuity, dissonance, and a few others (Josang et al., 2018). In particular, vacuity corresponds to the uncertainty mass of a subjective opinion $\omega$:

$$\text{vac}(\omega) = \mathcal{U}(\cdot|\mathbf{e}) = \frac{K}{S}, \quad S = \sum_{k=1}^{K}(e_k + 1) \tag{1}$$

Since vacuity measures lack of evidence (knowledge), it provides a natural means for exploration.

## 4 EVIDENTIAL CONSERVATIVE Q-LEARNING

**Overview.** We propose an Evidential Conservative Q-learning RS model to perform dynamic recommendations as shown in Figure 2. The model includes a sequential state encoder (SSE) to maintain a dynamic state space with a sliding window $W_t$ which moves along the user's interaction history $H_u$ over time to input new data into the SSE, and a conservative evidential-actor-critic (CEAC) module which functions as an RL agent to explore the item space by introducing the evidence-based uncertainty (*i.e.,* vacuity) into a new off-policy evidential RL setting. By incorporating previous state information, recent items captured by a sliding window, and the recommended items from the RL agent, the sequential encoder generates the current state $\mathbf{s}_t$. This state is further passed to the action network that predicts the mean and variance to form a Gaussian policy distribution. We sample a current action $\mathbf{a}_t$ from the policy distribution that corresponds to the latent preference of the user that simultaneously captures the past (via a previous state), current (through a sliding window) and future interest (through RL exploration). By leveraging the current action and total item embeddings from the Item Pool ($\mathcal{I}$), the evidence network provides the evidence that can be used to form the rating prediction for exploitation while estimating the uncertainty for better exploration. The Q-network (critic) generates a conservative evidential Q-value for conservative policy updates of the action network. Table 6 in Appendix A summarizes the major notations.

### 4.1 ENVIRONMENT SETUP

We start by describing the environment of the proposed evidential RL agent. The environment consists of a buffer containing all the user-interacted item embeddings with ground-truth ratings (*i.e.,*

the user's interaction history $H_u$), total item embeddings (*i.e.,* item pool $\mathcal{I}$), a sliding window $W_t$ that moves along the interaction history to generate new data input each time into the SSE for dynamic state maintenance, and a recommendation mechanism that specifies a score function to rank items and select top-$N$ of them to recommend. The score function encourages a balance between exploitation (based on predicted ratings) and exploration (based on evidence-based uncertainty) and is defined as:

$$\text{score}_{u,i} = \widehat{\text{rating}}_{u,i} + \frac{\lambda}{\log(h_i - W_t + 1)} \mathcal{U}_\pi(\cdot | \mathbf{e}_i) \tag{2}$$

where $i \in H_u, h_i > W_t$ with $h_i$ being the original appearance position of a recommended item $i$ in the user's interaction history and $W_t$ indexes the observation, *i.e.,* the last experienced item appearance position reached by the current sliding window, which separates the observed items from the unknown ones. Parameter $\lambda \in (0, 1)$ gradually balances the rating and the uncertainty along the training epochs and the test steps. $\widehat{\text{rating}}_{u,i}$ is the predicted rating for exploitation. Given $K$ possible rating classes, the evidence network (introduced later in this section) outputs an evidence vector $\mathbf{e}_i = (e_{i1}, ..., e_{iK})^\top$ for each item $i$. This will allow us to evaluate $\widehat{\text{rating}}_{u,i}$ as $\sum_{k=1}^{K} p_{ik} \times k$ where $p_{ik}$ is rating probability. Meanwhile, vacuity $\mathcal{U}_\pi(i | \mathbf{e}_i)$ for item $i$ can be evaluated through (1) for exploration. We aim to recommend unknown items that reflect the user's future interests while avoiding too risky recommendations far ahead of the current observation. To this end, we consider the original appearance position $h_i$ and down-weight the information value (captured by vacuity $\mathcal{U}_\pi(i | \mathbf{e}_i)$) of items that are interacted deep into the future. During testing, no down-weight is applied as $H_u, h_i$ are unavailable. Based on the ranking score, an RL agent will choose the top-$N$ items to form a list $\mathcal{N}_u$ and recommend them to the user. As feedback to the agent, the user provides the actual rating for each recommended item. Consequently, the *evidential reward* is

$$r_\pi^e(\mathbf{s}_t, \mathbf{a}_t) = \underbrace{\frac{1}{N} \sum_{i \in \mathcal{N}_u} (\text{rating}_{u,i} - \tau)}_{\mathbf{r}} + \underbrace{\lambda \frac{1}{N} \sum_{i \in \mathcal{N}_u} \mathcal{U}_\pi(\cdot | \mathbf{e}_i)}_{\mathbb{R}}. \tag{3}$$

where $\text{rating}_{u,i}$ is the user assigned rating, $\tau$ is the threshold to identify if a user provided rating is positive. Evidential reward aggregates the recommended items' rating as a traditional reward $\mathbf{r}$ balanced with their vacuity predictions as a measure of information gain, denoted as an uncertainty regularizer $\mathbb{R}$. During testing, for item $i'$ not appearing in user $u$'s interaction history $H_u$, a neutral rating $\text{rating}_{u,i'} = \tau$ will be assigned to give neutral feedback.

**Remark:** The novel use of vacuity, which is an evidence-based second-order uncertainty, for exploration in RL, can effectively identify uncertain and informative items (from a large item space), indicative of users' long-term interest. In particular, the proposed evidential reward encourages the RL agent to recommend items that the model has the least knowledge (as indicated by a high vacuity). After collecting the user feedback, the RL agent can most effectively gain knowledge of the user preference to make better recommendations in the long run. It should be noted that maximum entropy-based exploration, such as soft-actor-critic (SAC) (Haarnoja et al., 2018), may not reach an optimum policy. It has been shown that a high entropy may imply either high vacuity (lack of evidence) or high dissonance (conflict of strong evidence) (Shi et al., 2020). However, dissonance is not effective for exploration in RS due to its focus on confusing items mostly derived based on the users' current interests. We have also empirically shown this in a qualitative study by demonstrating better recommendation performance than SAC-based exploration in the experiments.

### 4.2 SEQUENTIAL STATE ENCODER

A specially designed sequential state encoder (SSE) is used to maintain the state space of a dynamic RS environment. A state $\mathbf{s}_t$ is generated by aggregating the previous state $\mathbf{s}_{t-1}$, items newly interacted by the user as the sliding window moves, and newly recommended items. By aggregating all this information, the current state can evolve from the previous state by a concatenated user embedding $\mathbf{u}_t$, which effectively captures the past preference and future predicted preference of the user. In particular, a sliding window $W_t$ of length $2N$ starts from the beginning of the user's interaction history ($H_u$) at time step 0, then it moves forward along the interaction history $N$ items each time as the newly observed items. The first $N$ already observed items in the moved window $W_{t+1}$ will be replaced by the top-$N$ recommendation list at last time step $t$, the rest will be the newly observed items in time step $t + 1$. Here, an *item* means an embedding vector that is generated by a pre-trained

Word2Vec network to encode item information from raw text descriptions. Then, $\mathbf{s}_t$ is formed by

$$\mathbf{s}_t = \text{SSE}(\mathbf{s}_{t-1}, \mathbf{u}_t). \tag{4}$$

We train the SSE by optimizing action $\mathbf{a}_t$ to maximize the conservative evidential Q-value in the critic network, which gives Equation (5), where $J_\pi(\phi, \omega)$ is the loss objective of the action network.

$$\nabla_\omega J_{\text{SSE}}(\omega) = \nabla_\omega J_\pi(\phi, \omega) \tag{5}$$

## 4.3 Conservative Evidential Actor Critic (CEAC)

CEAC performs conservative off-policy training using three key networks: *action network*, *critic network*, and *evidence network*, which will be detailed next.

**Conservative off-policy formulation.** RS models aim to leverage offline interactions to predict users' future interests. Since the interaction data is inherently sparse, it may train an RL model that overfits the limited training data, leading to overestimated Q-values of previously unseen interactions. To address the data scarcity and sparse rewards in a typical RS environment, we apply an off-policy learning scheme to promote the reuse of previously collected data and stabilize the training. To further prevent overestimation of the policy value, we extend the conservative Q-learning strategy developed for off-line RL settings (Kumar et al., 2020) and leverage it to penalize the target updated policy $\pi$'s Q-value estimation to avoid over-optimistic predictions on unknown items' ratings and information value as defined in (3). Meanwhile, we increase the Q-value estimate of the behavior policy $\pi_\beta$ that encodes the current knowledge of user interests to avoid the target policy from deviating too much from previous knowledge. These conservative regularizers are balanced with the traditional Bellman training objective using a hyper-parameter $\alpha$. In particular, we run the behavior policy in the RS environment $T$ time steps for one user $u$ as one RL episode. In each time step $t$, the RL model collects training tuples $(s_t, a_t, r_{\pi,t}^e, s_{t+1})$ into a replay buffer $D$. We iterate $M$ such RL episodes and then conduct a conservative evidential Q-value $Q_c^e$ evaluation of the currently learned policy by minimizing its Q-value while maximizing the Q value of the behavior policy, as given in (6), where $\theta$ is the parameter of critic network to minimize the loss objective $J_{Q_c^e}$:

$$J_{Q_c^e}(\theta) = \mathbb{E}_{(\mathbf{s}_t, \mathbf{a}_t, r_{\pi,t}^e, \mathbf{s}_{t+1}) \sim D} \left[ \frac{1}{2} (Q(\mathbf{s}_t, \mathbf{a}_t) - \hat{\mathcal{B}}^\pi Q(\mathbf{s}_t, \mathbf{a}_t))^2 \right] + \alpha \left( \mathbb{E}_{\mathbf{s}_t \sim D, \mathbf{a}_t \sim \pi(\cdot|\mathbf{s}_t)} [Q(\mathbf{s}_t, \mathbf{a}_t)] \right.$$
$$\left. - \mathbb{E}_{\mathbf{s}_t \sim D, \mathbf{a}_t \sim \pi_\beta(\cdot|\mathbf{s}_t)} [Q(\mathbf{s}_t, \mathbf{a}_t)] \right) \tag{6}$$

After conservative policy evaluation, we conduct a conservative policy improvement by optimizing the policy towards the optimal conservative evidential Q-value $Q_c^e$ objective as detailed in the action network. After policy improvement, we get a newly learned policy $\pi_{k+1}$ and again alternate between the policy evaluation and improvement steps until convergence. Once we get a converged policy, we update the previous behavior policy $\pi_\beta$ with the newly learned stable policy and begin the next $M$ RL episodes by collecting new training tuples into the replay buffer with the updated behavior policy. After iterating among all users in the training set, we call it an RL epoch. The detailed conservative off-policy formulation is shown in Algorithm 1 of Appendix D.

**Action network.** The action network (or policy network) utilizes the current state $\mathbf{s}_t$ from the offline replay buffer and outputs a policy distribution $\pi(\cdot|\mathbf{s}_t)$, which is modeled as a Gaussian. From this distribution, we sample an action $\mathbf{a}_t$ that is used in the evidence and the critic networks to provide recommendations or direct the policy update. According to our off-policy formulation, we use two separate action networks to represent the currently updated policy and previous behavior policy, respectively. The training of action network is given by

$$\nabla_\phi J_\pi(\phi) = (-\nabla_{\mathbf{a}_t} Q_c^e(\mathbf{s}_t, \mathbf{a}_t)) \times \nabla_\phi \pi(\mathbf{a}_t | \mathbf{s}_t, \phi) \tag{7}$$

where $\phi$ is the parameter of the action network to minimize the loss objective $J_\pi(\phi)$.

**Critic network.** The critic network is designed to approximate the conservative evidential Q-value by utilizing the current state $\mathbf{s}_t$ and action $\mathbf{a}_t$ in a fully connected neural network $Q_\theta(\mathbf{s}_t, \mathbf{a}_t)$. This Q-value judges whether the agent-generated actions match our training requirements. We derive an update formulation (6) for the critic network following the double DQN (Hasselt, 2010) that utilizes two critic networks to stabilize the training process, achieve faster convergence, and provide a better Q-value. Furthermore, the Q-network is optimized with stochastic gradient descent which back-propagates to the action network as well as the sequential encoder in an end-to-end fashion.

**Evidence network.** The evidence network predicts a Dirichlet distribution of class probabilities, which can be considered as an evidence-collection process. The learned evidence $\mathbf{e}_i = (e_{i1}, ..., e_{iK})^\top$ is informative to quantify the predictive uncertainty of recommended items. The network takes action $\mathbf{a}_t$ from the replay buffer and item pool $\mathcal{I}$ to provide class-level evidence. Then, the probability of rating $k$ is $p_{ik} = (e_{ik} + 1)/S_i$. To train the evidence network, we define a standard evidential loss (8) by utilizing the MSE loss between rating class probability $p_{ik}$ and the one-hot ground truth label $\mathbf{y}_i$, in which $y_{ik} = 1$ if $k$ is the correct rating, otherwise $y_{ik} = 0$:

$$J_{Evi}(\psi) = \sum_{i \in H_u} \sum_{k=1}^{K} (y_{ik} - p_{ik})^2 + \frac{p_{ik}(1 - p_{ik})}{S_i + 1} \tag{8}$$

We update the network by back-propagating the evidential loss $J_{Evi}(\psi)$ with its parameters $\psi$.

## 4.4 CONSERVATIVE EVIDENTIAL POLICY ITERATION

We first highlight some important concept differences resulting from our novel evidential conservative setting: 1) $\pi$ is actually an evidential policy governed by an evidential reward $r_\pi^e$, and 2) $\pi_\beta$ is the true behavior policy directly available from the off-policy setting without any simulation from offline data. Resulting of these two key symbol differences, the empirical Bellman operator $\hat{\mathcal{B}}^\pi$ should also be an evidential Bellman operator, where $\pi$ is an updated evidential policy. However, to leverage the existing theorems in relevant RL literature which are derived under the traditional RL concepts, we make the evidential terms explicit by restoring: 1) the evidential reward $r_\pi^e$ to $r + \mathcal{R}$, where $\mathcal{R}$ is the vacuity term in our case, and 2) evidential policy $\pi$ to non-evidential target policy $\bar{\pi}$. As a result, the empirical Bellman operator returns to its non-evidential version $\hat{\mathcal{B}}^{\bar{\pi}}$ and $\pi_\beta$ return back to non-evidence guided behavior policy $\bar{\pi}_\beta$. Based on these newly defined concepts, we first prove a conservative evidential policy evaluation with Q-value updated in (6). In Appendix C.2, we further show that such policy evaluation leads to a $\zeta$-safe policy improvement over the behavior policy. Then, we disclose its intrinsic relationship with an uncertainty-aware adjustment of optimism towards the previous behavior policy.

**Lemma 1** (Conservative Evidential Policy Evaluation). *Given a policy $\bar{\pi}$ and its conservative evidential Q value estimation $Q_c^e$ updated using (6), the expected conservative state value estimation $\hat{V}^{\bar{\pi}}(\mathbf{s})$ lower-bounds the actual state value $V^{\bar{\pi}}(\mathbf{s})$ for any state $s$ when the balancing factor*

$$\alpha \geq \frac{C_{r,T,\delta} R_{\max}}{1 - \gamma} \cdot \max_{\mathbf{s} \in D} \frac{1}{\sqrt{|D(\mathbf{s})|}} \left[ \sum_{\mathbf{a}} \frac{(\bar{\pi}(\mathbf{a}|\mathbf{s}) - \bar{\pi}_\beta(\mathbf{a}|\mathbf{s}))^2}{\bar{\pi}_\beta(\mathbf{a}|\mathbf{s})} \right]^{-1} \tag{9}$$

**Remark:** Due to the interaction between the evidence-based uncertainty in the evidential reward defined in (2), it leads to a more strict constraint (through the balancing parameter $\alpha$) in conservative policy evaluation to avoid risky recommendations when performing evidence based exploration. This is one critical difference from standard conservative Q-learning (Kumar et al., 2020), which does not consider evidential exploration. The detailed notation definition and proof are in Appendix C.1.

**Theorem 2** (Uncertainty Aware Optimism Adjustment). *The conservative evidential Q-value update in (6) has an intrinsic relationship with an uncertainty-aware adjustment of optimism towards the behavior policy by representing the evidential reward $r_\pi^e$ with a traditional reward $r$ added by an uncertainty regularizer $\mathbb{R}$, thus leading to a normal conservative Q-value update:*

$$\hat{Q}^{k+1} \leftarrow \min_{\hat{Q}^k} \quad \frac{1}{2} \mathbb{E}_{\mathbf{s},\mathbf{a},r,\mathbf{s}' \sim D} \left[ \left( \hat{Q}^k(\mathbf{a},\mathbf{s}) - \hat{\mathcal{B}}^{\bar{\pi}} \hat{Q}^k(\mathbf{a},\mathbf{s}) \right)^2 \right] \tag{10}$$
$$+ \alpha \left[ \mathbb{E}_{\mathbf{s} \sim D, \mathbf{a} \sim \bar{\pi}(\cdot|\mathbf{s})}[\hat{Q}^k(\mathbf{a},\mathbf{s})] - \left( \mathbb{E}_{\mathbf{s} \sim D, \mathbf{a} \sim \bar{\pi}_\beta(\cdot|\mathbf{s})}(1 + \mathbb{R})[\hat{Q}^k(\mathbf{a},\mathbf{s})] \right) \right] + const$$

**Remark:** The theorem manifests another novel interplay between the evidential uncertainty and the conservative policy update. From (10), we observe that the uncertainty regularizer $\mathbb{R}$ serves as an importance weight for the Q-value estimation of the behavior policy $\bar{\pi}_\beta$. This observation provides us with an elegant interpretation of the proposed ECQL algorithm: by adding the evidence-based uncertainty measure, we are actually adjusting the optimism of the previously learned behavior policy $\bar{\pi}_\beta$ according to the information gain (quantified by the vacuity) of a chosen action. If an action generated by a behavior policy leads to low information gain in the recommendation list (*i.e.,* low average vacuity), then the importance weight for the Q-value of such behavior policy is lower and the ECQL algorithm will discourage the RL agent to keep imitating the behavior policy. We leave the detailed derivation in Appendix C.3.

Table 2: Recommendation performance comparison (average precision P@N, discounted cumulative gain nDCG@N and test return R@T)

| Category | Model | MovieLens-1M | | | MovieLens-100K | | | Netflix | | | Yahoo! Music | | |
|---|---|---|---|---|---|---|---|---|---|---|---|---|---|
| | | P@5 | nDCG@5 | R@16 | P@5 | nDCG@5 | R@10 | P@5 | nDCG@5 | R@16 | P@5 | nDCG@5 | R@10 |
| Sequential | CASER | 0.5762 | 0.4613 | N/A | 0.5434 | 0.4428 | N/A | 0.5633 | 0.4532 | N/A | 0.5745 | 0.4315 | N/A |
| | SASRec | 0.6058 | 0.4862 | N/A | 0.5624 | 0.4515 | N/A | 0.5958 | 0.4621 | N/A | 0.5826 | 0.4422 | N/A |
| | ADFM | 0.5749 | 0.4642 | N/A | 0.5387 | 0.4407 | N/A | 0.5695 | 0.4412 | N/A | 0.5541 | 0.4218 | N/A |
| | SDIM | 0.5877 | 0.4658 | N/A | 0.5471 | 0.4463 | N/A | 0.5739 | 0.4498 | N/A | 0.5638 | 0.4314 | N/A |
| | HPMN | 0.5962 | 0.4745 | N/A | 0.5589 | 0.4502 | N/A | 0.5862 | 0.4574 | N/A | 0.5755 | 0.4394 | N/A |
| | BERT4Rec | 0.6122 | 0.4957 | N/A | 0.5834 | 0.4855 | N/A | 0.5996 | 0.4667 | N/A | 0.5901 | 0.4522 | N/A |
| | Seq2Seq | 0.5818 | 0.4752 | N/A | 0.5579 | 0.4614 | N/A | 0.5648 | 0.4554 | N/A | 0.5762 | 0.4332 | N/A |
| | $S^3$-Rec | 0.6108 | 0.4926 | N/A | 0.5792 | 0.4767 | N/A | 0.5884 | 0.4602 | N/A | 0.5786 | 0.4358 | N/A |
| | CL4SRec | 0.6135 | 0.4952 | N/A | 0.5813 | 0.4781 | N/A | 0.5902 | 0.4688 | N/A | 0.5841 | 0.4423 | N/A |
| Dynamic | timeSVD++ | 0.5341 | 0.4328 | N/A | 0.5034 | 0.4145 | N/A | 0.4938 | 0.4082 | N/A | 0.4838 | 0.3874 | N/A |
| | CKF | 0.5567 | 0.4481 | N/A | 0.5285 | 0.4322 | N/A | 0.5144 | 0.4280 | N/A | 0.5078 | 0.4155 | N/A |
| Deep Learning | DeepFM | 0.5647 | 0.4625 | N/A | 0.5428 | 0.4514 | N/A | 0.5315 | 0.4434 | N/A | 0.5217 | 0.4324 | N/A |
| | DCNv2 | 0.6152 | 0.5187 | N/A | 0.6158 | 0.5166 | N/A | 0.5784 | 0.4905 | N/A | 0.5715 | 0.4799 | N/A |
| Bandit | Lin | 0.6191 | 0.5205 | 58.26 | 0.6174 | 0.5271 | 35.12 | 0.5842 | 0.5082 | 57.15 | 0.5849 | 0.4876 | 33.81 |
| | HLin | 0.6088 | 0.5205 | 58.25 | 0.6172 | 0.5270 | 35.08 | 0.5819 | 0.5047 | 57.09 | 0.5813 | 0.4855 | 33.62 |
| | CoLin | 0.6212 | 0.5236 | 58.87 | 0.6227 | 0.5295 | 35.49 | 0.5875 | 0.5096 | 58.13 | 0.5875 | 0.4894 | 34.25 |
| Reinforce | $\epsilon$-greedy | 0.5977 | 0.4834 | 55.15 | 0.5580 | 0.4556 | 32.74 | 0.5850 | 0.4765 | 55.64 | 0.5909 | 0.4812 | 31.57 |
| | SAC | 0.6105 | 0.5215 | 61.63 | 0.6208 | 0.5277 | 35.78 | 0.5942 | 0.4839 | 57.82 | 0.6074 | 0.5075 | 35.44 |
| | DRN | 0.6057 | 0.5199 | 55.26 | 0.6154 | 0.5268 | 32.78 | 0.5826 | 0.4720 | 57.58 | 0.6085 | 0.5121 | 33.15 |
| | LIRD | 0.6238 | 0.5332 | 61.85 | 0.6137 | 0.5222 | 35.88 | 0.6134 | 0.5214 | 63.59 | 0.6193 | 0.5238 | 38.19 |
| | SAR | 0.5938 | 0.4956 | 60.95 | 0.5957 | 0.4968 | 35.05 | 0.5892 | 0.5026 | 62.84 | 0.5952 | 0.5014 | 37.26 |
| | REINforCE | 0.6160 | 0.5285 | 62.24 | 0.6104 | 0.5138 | 37.66 | 0.6134 | 0.5294 | 64.25 | 0.6174 | 0.5238 | 39.76 |
| | ResAct | 0.6188 | 0.5215 | 62.34 | 0.6195 | 0.5234 | 38.96 | 0.6175 | 0.5319 | 64.67 | 0.6215 | 0.5288 | 39.94 |
| | HAC | 0.5989 | 0.5005 | 61.28 | 0.6021 | 0.5098 | 38.55 | 0.5966 | 0.5138 | 63.14 | 0.6009 | 0.5054 | 38.18 |
| | SAS-SQN | 0.5682 | 0.4755 | 57.69 | 0.5711 | 0.4795 | 34.89 | 0.5704 | 0.4852 | 60.32 | 0.5765 | 0.4735 | 34.96 |
| | **ECQL** | **0.6313** | **0.5365** | **68.54** | **0.6379** | **0.5386** | **42.31** | **0.6336** | **0.5372** | **70.19** | **0.6232** | **0.5330** | **43.59** |

## 5 EXPERIMENTS

We conduct experiments on multiple real-world datasets: *Movielens-1M*, *Movielens-100K*, *Netflix*, and *Yahoo! Music*. For baselines, we use **dynamic** models: timeSVD++ (Koren, 2009), CKF (Gultekin & Paisley, 2014); **sequential** models: CASER (Tang & Wang, 2018), SASRec (Kang & McAuley, 2018), ADFM (Li et al., 2022),SDIM (Cao et al., 2022),HPMN (Ren et al., 2019), BERT4Rec (Sun et al., 2019), Seq2Seq (Ma et al., 2020), $S^3$-Rec (Zhou et al., 2020), CL4SRec (Xie et al., 2022); **Deep Learning** models: DeepFM (Guo et al., 2017), DCNv2 (Wang et al., 2021); **Bandit** models: Lin (Li et al., 2010a), HLin (Wang et al., 2016), CoLin (Wu et al., 2016); and **RL-based** models: $\epsilon$-greedy (Zhao et al., 2013), SAC (Haarnoja et al., 2018), DRN (Zheng et al., 2018), LIRD (Zhao et al., 2017), SAR (Antaris & Rafailidis, 2021), REINforCE (Chen et al., 2019), ResAct (Xue et al., 2022), HAC (Liu et al., 2023), SAS-SQN (Silva et al., 2024). Further details about datasets, metrics, settings, and baselines are in Appendix E.

### 5.1 EVALUATION METRICS

We use three standard metrics: **Precision@N** and **nDCG@N** to measure the average recommendation performance across all time steps. We also use test return **R@T** as the sum of the average rating of all recommended items until time step $T$ for the RL-based methods comparison. For any particular test user, **Positive Count**, and **Genre Count** are directly applied to measure the qualitative performance. We also consider a standard offline evaluation metric *Normalized Capped Importance Sampling* (**NCIS**) (Swaminathan & Joachims, 2015) for RL comparison.

- **Precision@N**: It is the fraction of the top-$N$ items recommended in each step of the episode that are positive (*i.e.,* rating $> \tau$) to the user. We average overall test users as the final precision.
- **nDCG@N**: Normalized Discounted Cumulative Gain (nDCG@N) measures ranking quality, considering the top-$N$ recommended items' information gain normalized by top-$N$ items of the ideal ranking list based on the ordered ground truth rating in each step of the RL episode. We average overall test users as the final precision.
- **Test Return**: We evaluate test rewards based on the average ground-truth ratings of top-$N$ recommended items in each step within an RL episode. Test Return considers the sum of the test rewards before any step within an RL episode and averages across all test users.
- **Positive Count**: It is the total count of the top-$N$ items recommended in each step of the episode that are positive (*i.e.,* rating $> \tau$) to a particular test user.
- **Genre Count**: It is the total genre count of the top-$N$ items recommended in each step of the episode to a particular test user.
- **NCIS**: It is a variant of average cumulative reward, weighted by the normalized capped importance sampling score based on the policy probability, formulated in E.

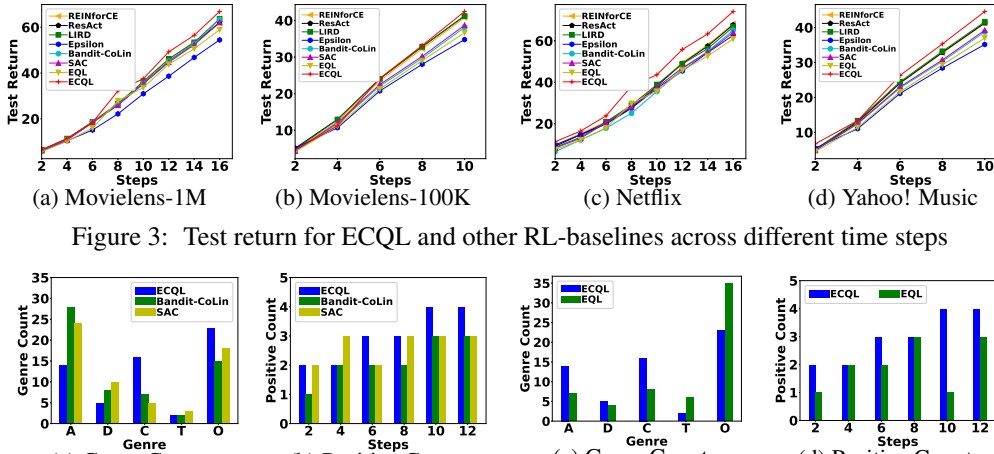

Figure 3: Test return for ECQL and other RL-baselines across different time steps

Figure 4: Genre and Positive Count comparisons with RL models using bandit-based (CoLin) and entropy-based (SAC) explorations (a-b) as well as EQL without conservative learning (c-d)

## 5.2 RECOMMENDATION PERFORMANCE COMPARISON

Table 2 summarizes the recommendation performance from all models. The proposed model benefits from both the sequential state encoder and CEAC module so that it provides better results in all datasets. Sequential models achieve less ideal performance due to their focus on short-term user preference, which is less effective in capturing long-term interest. RL methods have shown a clear advantage due to their focus on maximizing expected long-term rewards. By leveraging the vacuity guided exploration, ECQL achieves the best performance among all RL based models.

To further demonstrate ECQL's superior performance, we conduct an RL baseline comparison on an offline evaluation metric NCIS, which measures the normalized capped importance sampling score weighted cumulative reward along the whole trajectory across all test users. As Table 3 shows, ECQL achieves significant better performance, *i.e.,* higher normalized average cumu-

Table 3: RL comparison on offline evaluation metric Normalized Capped Importance Sampling (NCIS).

| Model | MovieLens-1M | MovieLens-100K | Netflix | Yahoo! Music |
|---|---|---|---|---|
| $\epsilon$-greedy | 3.088 | 3.029 | 3.154 | 3.135 |
| SAC | 3.248 | 3.165 | 3.324 | 3.258 |
| DRN | 3.035 | 2.983 | 3.102 | 3.084 |
| LIRD | 3.545 | 3.478 | 3.629 | 3.569 |
| SAR | 3.835 | 3.765 | 3.928 | 3.843 |
| REINforCE | 3.926 | 3.845 | 4.024 | 3.939 |
| ResAct | 4.035 | 3.980 | 4.126 | 4.083 |
| HAC | 3.852 | 3.784 | 3.946 | 3.891 |
| SAS-SQN | 3.659 | 3.598 | 3.754 | 3.690 |
| **ECQL** | **4.124** | **4.112** | **4.223** | **4.178** |

lative reward than all the RL competitors. As can be seen, the test return for ECQL and the RL-based baselines, including DRN, LIDR, and CoLin (other models will be discussed in the ablation study), are quite close in the initial steps. But in later steps, ECQL clearly outperforms the other baselines. This is because the model is able to explore more effectively during the training process to enhance the knowledge of the model. The effectiveness of vacuity guided exploration will be further investigated in our ablation study as detailed next.

## 5.3 ABLATION STUDY

We first investigate the effect of exploration of ECQL by comparing with two alternative designs: (1) w/o vacuity (denoted as **Epsilon**) and (2) w/o conservative learning guided exploration (denoted as **EQL**). We also compare exploration using the first-order uncertainty, employed by soft-actor-critic (SAC). Figure 3 shows that without uncertainty-guided exploration, Epsilon collects the least test return in the long run. SAC utilizes entropy-based exploration and achieves better test return than without uncertainty-guided exploration, almost equal to the performance of strongest reinforced baselines LIRD, REINforCE, and ResAct, across four different data sets. This provides evidence that the role of the exploration is crucial in RL-based recommendations. However, its performance is worse than the vacuity-based ECQL method. This is because vacuity-guided exploration allows our model to explore the most informative items so that it can gain the most knowledge to form an optimal policy. However, given the sparse rewards, it is essential that the recommendation does not

Table 4: Recommended movies for an example user

| Model | Movies | Movie Genre | Vacuity |
|-------|--------|-------------|---------|
| | Kids of the Round Table (1995) | **Adventure,Fantasy** | 0.12 |
| | Postino, Il (The Postman) (1994) | **Adventure, Romance** | 0.11 |
| ECQL | How to Make an American Quilt (1995) | **Drama** | 0.14 |
| | Pocahontas (1995) | **Musical** | 0.12 |
| | Three Lives and Only One Death (1996) | Comedy | 0.22 |
| | Karate Kid, Part II, The (1986) | **Fantasy** | 0.07 |
| | Return of the Pink Panther, The (1974) | Comedy | 0.12 |
| SAC | Lawnmower Man 2: Beyond Cyberspace (1996) | Sci-Fi,Thriller | 0.09 |
| | Young Sherlock Holmes (1985) | **Adventure** | 0.11 |
| | Love in Bloom (1935) | **Romance** | 0.09 |
| | Ruling Class, The (1972) | Tragedy | 0.14 |
| | Private Benjamin (1980) | Comedy | 0.13 |
| EQL | Mighty Joe Young (1998) | **Adventure** | 0.08 |
| | Christmas Vacation (1989) | Action | 0.14 |
| | Father of the Bride Part II (1995) | Sci-Fi | 0.14 |

deviate too much from user's interest. As shown, EQL performs worse than ECQL by a clear margin, especially in the later steps when the model gets richer knowledge about the user through interactions.

We further evaluate the impact of the two key modules of ECQL: SSE and CEAC. We report P@5 and nDCG@5 on Movielens-1M in Table 5. The integration of the two modules significantly outperforms each individual module. Additional ablation studies are presented in the Appendix.

Table 5: Impact of each module

| SSE | CEAC | P@5 | nDCG@5 |
|-----|------|-----|--------|
| ✓ | ✗ | 0.4899 | 0.4175 |
| ✗ | ✓ | 0.6044 | 0.5108 |
| ✓ | ✓ | 0.6313 | 0.5365 |

### 5.4 QUALITATIVE STUDY

**Impact of vacuity for exploration.** We conduct a qualitative analysis to show the advantage of using evidence-based uncertainty (*i.e.,* vacuity) for RL exploration when compared with other two competitive baselines: entropy-guided exploration as in the soft actor-critic (SAC) and a contextual bandit algorithm (CoLin). We select a random test user from the Movielens-1M dataset and show the genre counts and positive counts of recommended items in Figure 4 (a) and (b). At the initial few steps, SAC has more positive counts but is less effective in exploration. The proposed ECQL is able to explore more informative items (evidenced by more diverse genres). In later steps, ECQL consistently outperforms both SAC and CoLin in positive counts due to the better utilization of evidence-based uncertainty to discover more informative future items that could reflect the user's long-term preference, as shown in Figure 4 (b). Table 4 shows the predicted vacuity for each recommended item in step 10. The overall higher vacuity scores indicate that ECQL recommends more items that are currently unknown to the users, which is instrumental in exploring their long-term interests. ECQL also explores more diverse genres and identifies four out of five important items (genre types in bold) that are positive in ground-truth ratings. Benefiting from better exploration, ECQL eventually achieves a much better test return compared to SAC and CoLin as shown in Figure 3. More detailed discussions about how to control the exploration through hyper-parameter $\lambda$ are included in Appendix E.4.

**Impact of conservative learning.** The last two plots in Figure 4 show that comparing to EQL, which lacks a conservative constraint, our proposed model achieves much higher positive counts across different steps while exploring diverse genres. Because of the sparse reward space and limited training data, making reasonable and safer recommendations based on current confirmed knowledge is important to maintain a user base. As shown in Table 4, EQL explores a wide range of different types of movies, but only one of them is positive reflecting user's true interest.

## 6 CONCLUSION

In this paper, we propose a novel evidential conservative Q-learning framework for dynamic recommendations. The proposed ECQL framework learns an effective and safe recommendation policy by integrating both the evidence-based exploration and conservative learning. ECQL integrates a customized SSE to generate the current state that accurately captures user interest and a conservative evidential-actor-critic module which functions as an RL agent to perform evidence-based exploration and trained through conservative evidential Q-learning in an off-policy formulation. We theoretically prove that the conservative evidential Q-value provides an uncertainty-aware adjustment of optimism towards the behavior policy and lower-bounds the actual value of the target policy.

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

# Appendix

## Table of Contents

**Organization of Appendix.** In this Appendix, we first summarize the major mathematical notations in Appendix A. We give an overview of representative static recommendation models in Appendix B. We then present the proofs of lemmas and theorems in Appendix C. We show the detailed ECQL algorithm in Appendix D. We present the details of the datasets, experimental setting and baseline models in Appendix E. Further, we include some additional comparison results in Appendix E.3 and conduct an ablation study as well as a qualitative analysis in Appendix E.4 and E.5. Limitation and future extension of our work are discussed in Appendix F. The link to the source code is given in Appendix G.

## A   SUMMARY OF NOTATIONS

Table 6: Summary of notations

| Symbol Group | Notation | Description |
|---|---|---|
| ECQL | $T, M$ | step size or length of episode and episode size for ECQL training |
| | $u, i, h_i$ | user (episode), item and item's position indices |
| | $\mathbf{u}_t$ | user $u$'s embedding at time step t |
| | $\mathbf{s}_t, \mathbf{a}_t$ | state and action at time t |
| | $e_{ik}, p_{ik}$ | evidence and evidence-based probability on rating class $k$ for item $i$ |
| | $y_i$ | one-hot rating label on item $i$ |
| | $\phi, \psi$ | parameters of action and evidence networks |
| | $\theta, \omega$ | parameters of critic and SSE networks |
| | $\pi, Q(\mathbf{s}_t, \mathbf{a}_t)$ | recommendation policy or action network, Q value function or critic network |
| | $\widehat{\text{rating}}_{u,i}, \text{rating}_{u,i}$ | predicted and actual rating for user $u$ on item $i$ |
| | $\text{score}_{u,i}$ | evidential score for user $u$ on item $i$ |
| | $\tau, \lambda$ | rating threshold and balance hyper-parameter for exploitation and exploration |
| | $\gamma$ | RL discount factor for test return |
| | $\mathcal{U}_\pi(\cdot \| \mathbf{e}_i)$ | item $i$'s evidence-based uncertainty |
| | $r_\pi^e(\mathbf{s}_t, \mathbf{a}_t), Q^e(\mathbf{s}_t, \mathbf{a}_t)$ | evidential reward and Q value |
| Environment | $\mathcal{P}(\mathbf{s}'\|\mathbf{s}, \mathbf{a})$ | state transition probability |
| | $K$ | the number of rating class |
| | $W^l, N$ | the number of interacted items in the sliding window and recommended items in the recommendation list in each time step |
| | $\mathcal{N}_u$ | top-$N$ recommended items list for user $u$ |
| | $H_u, \mathcal{I}$ | user $u$'s interaction history and item pool |
| Theoretical Results | $C_{r,T,\delta}, C_{r,\delta}, C_{T,\delta}$ | constants dependent on the concentration properties (variance) of evidential reward $r_\pi^e(\mathbf{a}, \mathbf{s})$ and/or state transition matrix $T(\mathbf{s}'\|\mathbf{s}, \mathbf{a})$ |
| | $\alpha, \delta, \zeta$ | hyper-parameters controlling conservatism, high probability and safe policy improvement guarantee. |
| | $J(\bar{\pi}, \hat{M}), J(\bar{\pi}, M)$ | discounted return of a policy $\bar{\pi}$ in the empirical MDP, $\hat{M}$ and actual MDP, M. |
| | $\|D\|, \|A\|, d_{\hat{M}}^{\bar{\pi}}$ | magnitude of state and action space, state distribution from empirical MDP $\hat{M}$(offline dataset) |
| | $D_{ECQL}(\bar{\pi}, \bar{\pi}_\beta)$ | ECQL regularizer detailed in C.1. |

## B   ADDITIONAL RELATED WORK

To provide a complete review of the existing literature on recommender systems, this section gives an overview of representative static and dynamic recommendation models that complement the sequential and RL-based models discussed in the main paper. We also discuss some other RL exploration strategies and use of uncertainty in recommender systems.

**Static recommendation models.** Matrix Factorization (MF) leverages user and item latent factors to infer user preferences (Koren et al., 2009; Funk, 2006; Koren, 2008). MF is further extended with Bayesian Personalized Ranking (BPR) (Rendle et al., 2012) and Factorization Machine (FM) (Rendle, 2010). Recently, deep learning-based recommender systems (Cheng et al., 2016; Guo et al., 2017) have achieved impressive performance. DeepFM (Guo et al., 2017) integrates traditional FM and deep learning to learn low- and high-order feature interactions. The wide and deep networks are jointly trained in (Cheng et al., 2016) for better memorization and generalization. In graph-based methods (Berg et al., 2017), users and items are represented as a bipartite graph and links are predicted to provide recommendations. Similarly, Neural Graph Collaborative Filtering (Wang et al., 2019) explicitly encodes the collaborative signal via high-order connectivities in the user-item bipartite graph via embedding propagation.

**Dynamic recommendation models.** Dynamic recommendation models shift users' latent preferences over time to incorporate temporal information. TimeSVD++ (Koren, 2009) considers time-specific factors, which uses additive bias to model user and item-related temporal changes. Gaussian state-space models have been used to introduce time-evolving factors with a one-way Kalman filter (Gultekin & Paisley, 2014). Hidden Markov model (Sahoo et al., 2012) and Poisson emission (Charlin et al., 2015) have also been leveraged to model temporal dependencies. However, these models primarily capture users' evolving preferences from the past that are less effective to predict future interactions.

**Additional RL exploration strategy.** ICM Pathak et al. (2017) and RND Burda et al. (2018) are designed for complex deep reinforcement learning (DRL) environments (e.g., Atari games) with sparse reward signals, where exploring novel states is critical to uncover rare rewards, penalizing events, or turning points. ICM uses curiosity to facilitate exploration, measured by the error in predicting the next state from the current state-action pair, leveraging a forward dynamic model trained on prior interactions. Additionally, it employs an inverse dynamic model to filter out irrelevant state representations. Similarly, RND employs a fixed, randomly initialized network (random distillation network) as a reference model. A predictor neural network is trained to replicate the outputs of the reference model, and the prediction error for any new state serves as a novelty score. However, directly applying these exploration strategies to dynamic recommendation settings may be less effective, as they are primarily designed for high-dimensional DRL environments, such as Atari games. These methods are not tailored to accurately quantify rating prediction uncertainty, which is crucial for encouraging the exploration of novel items in recommendation systems.

**Use of uncertainty in recommendation models.** Quantifying the uncertainty of a recommender system (RS) Cleger-Tamayo et al. (2013); McNee et al. (2003) is important because it helps to reveal which rating predictions or recommended items are likely to be wrong Bobadilla et al. (2018). Uncertainty estimates can improve an RS in three main ways:

1. **Convey recommendation confidence** Cleger-Tamayo et al. (2013); McNee et al. (2003). Displaying uncertainty alongside recommendations can enhance the trustworthiness and credibility of a RS. For instance, when the RS is uncertain about a recommendation, providing an explanation can help users understand the rationale behind the suggestion. This approach reduces the risk of confusing or untrustworthy recommendations and improves the overall user experience.
2. **Guide which items to recommend** Mazurowski (2013); Ortega et al. (2021); Bouneffouf et al. (2013). Typically, a recommender system (RS) suggests items it predicts to be most relevant to the user. By incorporating uncertainty estimates, the RS can avoid recommending items with high predicted relevance but significant uncertainty, thereby reducing the risk of poor recommendations. Alternatively, the RS could strategically include uncertain recommendations to encourage user discovery of new items and gather valuable feedback for improving future predictions.
3. **Optimize recommendation process** Burke (2002); da Costa et al. (2018). Uncertainty can also assist a recommender system (RS) in optimizing its internal processes. For example, the CoRec system leverages uncertainty to determine which predictions to include in its training data. More broadly, uncertainty estimates can guide the RS in selecting between different models or strategies. They are also valuable in active learning scenarios, where the system solicits user feedback on items it is uncertain about, thereby improving the quality of future recommendations.

## C   Proofs of Theoretical Results

In this section, we provide detailed proofs of Lemma 1, Theorem 2, and $\zeta$-safe policy improvement guarantees over behavior policy by iterating conservative policy evaluation (6) and update (7).

### C.1   Proof of Lemma 1

**Proof overview.**   We first introduce some important notations that will be used in the proof of the Lemma. Our proof shows that conservative Q-value estimation updated step by step using (12) leads to a lower-bounded state-marginal value estimation compared to its previous iterate. Then, we compute the fixed point of such iteration to conclude that the converged state value estimation $\hat{V}^{\bar{\pi}}(\mathbf{s})$ lower-bounds the actual state value $V^{\bar{\pi}}(\mathbf{s})$. We first conduct the whole derivation in tabular Q-learning setting where state transition is finite and precise (Auer & Ortner, 2006), then we incorporate the sampling error to make it a complete RL process.

**Notations.**   Let $k \in \mathbb{N}$ denote an Q-iteration of policy evaluation. In an iteration $k$, the objective in (6) is optimized using the Bellman backup (*i.e.*, $\hat{\mathcal{B}}^{\bar{\pi}}\hat{Q}^{k-1}$) as the target value which is given as Equation (11) by following a double DQN training fashion, where $\mathcal{R}$ is the evidential regularizer, $\gamma$ is the discounted factor, $\tilde{Q}$ is the target Q-network representing a stable evaluation and $\hat{Q}$ is the Q-network representing the current learned policy's Q-value. Let $Q^k$ denote the true, tabular Q-function iterate in the actual MDP $M$, and $Q^k$ is estimated using the empirical Bellman operator $\hat{\mathcal{B}}$ as: $Q^k \leftarrow \hat{\mathcal{B}}^{\bar{\pi}}\hat{Q}^k$ (for policy evaluation). We redefine $Q_c^e$ in the main paper as $Q^k$ when $k \to \infty$ (*i.e.*, the $k$-th Q-function iterate reaches convergence). We emphasize that $\bar{\pi}_\beta$ is the actual behavior policy directly available from last policy update, $\alpha$ is the hyper-parameter to guarantee conservatism, and $\hat{V}^k$ denotes the state value estimation, which is the expectation of corresponding Q-value under a given policy $\bar{\pi}$: $\hat{V}^k := \mathbb{E}_{\mathbf{a} \sim \bar{\pi}(\mathbf{a}|\mathbf{s})}[\hat{Q}^k(\mathbf{a}, \mathbf{s})]$.

$$
\begin{aligned}
\hat{\mathcal{B}}^{\bar{\pi}}\hat{Q}^{k-1}(\mathbf{s}_t, \mathbf{a}_t) = \mathbb{E}_{\mathbf{s}_t, \mathbf{a}_t, \mathbf{s}_{t+1} \sim D, \mathbf{a}_{t+1} \sim \bar{\pi}}[(r(\mathbf{s}_t, \mathbf{a}_t) \\
+ \gamma \times \min\{\tilde{Q}^{k-1}(\mathbf{s}_{t+1}, \mathbf{a}_{t+1}), \hat{Q}^{k-1}(\mathbf{s}_{t+1}, \mathbf{a}_{t+1})\})].
\end{aligned} \tag{11}
$$

**Assumptions.**   Following (Osband et al., 2016; O' Donoghue, 2021; Auer et al., 2008), we use concentration properties of $\hat{\mathcal{B}}^{\bar{\pi}}$ to control the sampling error. Formally, for all $\mathbf{s}, \mathbf{a} \in D$, with probability $\geq 1 - \delta$, there exists a constant $C_{r,T,\delta}$ dependent on the concentration properties (variance) of reward $r(\mathbf{s}, \mathbf{a})$ and state transition matrix $T(\mathbf{s}'|\mathbf{s}, \mathbf{a})$ that satisfies $|\hat{\mathcal{B}}^{\bar{\pi}} - \mathcal{B}^{\bar{\pi}}|(\mathbf{s}, \mathbf{a}) \leq \frac{C_{r,T,\delta}}{|\sqrt{D(\mathbf{s}, \mathbf{a})}|}$, where $\delta \in (0, 1)$. $\frac{1}{|\sqrt{D(\mathbf{s}, \mathbf{a})}|}$ denotes a vector of size $|S||A|$ containing square root inverse counts for each state-action pair, except when $D(\mathbf{s}, \mathbf{a}) = 0$, in which case the corresponding entry is a very large but finite value $\delta \geq \frac{2R_{\max}}{1-\gamma}$.

**Proof.**   In the tabular setting, we can set the derivative of the modified objective in (6) to 0, and compute the Q-function update induced in the exact, tabular setting, thus giving (12) as one optimization step (this assumes $\hat{\mathcal{B}}^{\bar{\pi}} = \mathcal{B}^{\bar{\pi}}$).

$$
\forall \mathbf{s}, \mathbf{a} \quad \hat{Q}^k(\mathbf{a}, \mathbf{s}) = \mathcal{B}^{\bar{\pi}}\hat{Q}^{k-1}(\mathbf{a}, \mathbf{s}) + \tilde{\mathcal{R}} - \alpha\left[\frac{\bar{\pi}(\mathbf{a}|\mathbf{s})}{\bar{\pi}_\beta(\mathbf{a}|\mathbf{s})} - 1\right]. \tag{12}
$$

Where $\tilde{\mathcal{R}}$ is the expectation of the evidential regularizer under the off-policy state-action distribution $D$. Note that for state-action pairs $(\mathbf{a}, \mathbf{s})$ such that, $\bar{\pi}(\mathbf{a}|\mathbf{s}) < \bar{\pi}_\beta(\mathbf{a}|\mathbf{s})$, we are in fact adding a positive quantity, $1 - \frac{\bar{\pi}(\mathbf{a}|\mathbf{s})}{\bar{\pi}_\beta(\mathbf{a}|\mathbf{s})}$. To the Q-function obtained, a point-wise lower bound is not guaranteed, *i.e.*, $\forall \mathbf{s}, \mathbf{a}, s.t. \hat{Q}^k(\mathbf{a}, \mathbf{s}) \leq \hat{Q}^{k-1}(\mathbf{a}, \mathbf{s})$. However, we show that on the other hand the expected state value of the estimated Q-function, *i.e.*, $\hat{V}^k$ always lower-bounds its previous iterate, since:

$$
\hat{V}^k(\mathbf{s}) := \mathbb{E}_{\mathbf{a} \sim \bar{\pi}(\mathbf{a}|\mathbf{s})}\left[\hat{Q}^k(\mathbf{a}, \mathbf{s})\right] = \mathcal{B}^{\bar{\pi}}\hat{V}^{k-1}(\mathbf{s}) - \alpha\mathbb{E}_{\mathbf{a} \sim \bar{\pi}(\mathbf{a}|\mathbf{s})}\left[\frac{\bar{\pi}(\mathbf{a}|\mathbf{s})}{\bar{\pi}_\beta(\mathbf{a}|\mathbf{s})} - (1 + \frac{\tilde{\mathcal{R}}}{\alpha})\right]. \tag{13}
$$

The value of the policy $\hat{V}^k$ is underestimated comparing to $\hat{V}^{k-1}$, since we can show that the evidential term $D_{ECQL}(\mathbf{s}) := \sum_{\mathbf{a}} \bar{\pi}(\mathbf{a}|\mathbf{s}) \left[ \frac{\bar{\pi}(\mathbf{a}|\mathbf{s})}{\bar{\pi}_\beta(\mathbf{a}|\mathbf{s})} - (1 + \frac{\tilde{\mathcal{R}}}{\alpha}) \right]$ is always positive under the condition that $\alpha \geq [\sum_{\mathbf{a}} \frac{(\bar{\pi}(\mathbf{a}|\mathbf{s}) - \bar{\pi}_\beta(\mathbf{a}|\mathbf{s}))^2}{\bar{\pi}_\beta(\mathbf{a}|\mathbf{s})}]^{-1}$, which is the evidential constraint on conservative parameter $\alpha$ to guarantee underestimation in each value update. To note this, we present the following derivation:

$$
\begin{aligned}
D_{ECQL}(\mathbf{s}) &:= \sum_{\mathbf{a}} \bar{\pi}(\mathbf{a}|\mathbf{s}) \left[ \frac{\bar{\pi}(\mathbf{a}|\mathbf{s})}{\bar{\pi}_\beta(\mathbf{a}|\mathbf{s})} - (1 + \frac{\tilde{\mathcal{R}}}{\alpha}) \right] \\
&= \sum_{\mathbf{a}} (\bar{\pi}(\mathbf{a}|\mathbf{s}) - \bar{\pi}_\beta(\mathbf{a}|\mathbf{s}) + \bar{\pi}_\beta(\mathbf{a}|\mathbf{s})) \left[ \frac{\bar{\pi}(\mathbf{a}|\mathbf{s})}{\bar{\pi}_\beta(\mathbf{a}|\mathbf{s})} - (1 + \frac{\tilde{\mathcal{R}}}{\alpha}) \right] \\
&= \sum_{a} (\bar{\pi}(\mathbf{a}|\mathbf{s}) - \bar{\pi}_\beta(\mathbf{a}|\mathbf{s})) \left[ \frac{\bar{\pi}(\mathbf{a}|\mathbf{s}) - \bar{\pi}_\beta(\mathbf{a}|\mathbf{s})}{\bar{\pi}_\beta(\mathbf{a}|\mathbf{s})} \right] + \sum_{\mathbf{a}} \bar{\pi}_\beta(\mathbf{a}|\mathbf{s}) \left[ \frac{\bar{\pi}(\mathbf{a}|\mathbf{s})}{\bar{\pi}_\beta(\mathbf{a}|\mathbf{s})} - (1 + \frac{\tilde{\mathcal{R}}}{\alpha}) \right] \\
&= \sum_{\mathbf{a}} \underbrace{\left[ \frac{(\bar{\pi}(\mathbf{a}|\mathbf{s}) - \bar{\pi}_\beta(\mathbf{a}|\mathbf{s}))^2}{\bar{\pi}_\beta(\mathbf{a}|\mathbf{s})} \right]}_{\geq 0} - \frac{\tilde{\mathcal{R}}}{\alpha} \geq 0, \text{(R is an uncertainty regularity between 0 to 1)}
\end{aligned}
$$

which requires $\alpha \geq \dfrac{1}{\left[ \sum_{\mathbf{a}} \frac{(\bar{\pi}(\mathbf{a}|\mathbf{s}) - \bar{\pi}_\beta(\mathbf{a}|\mathbf{s}))^2}{\bar{\pi}_\beta(\mathbf{a}|\mathbf{s})} \right]} \geq \dfrac{\tilde{\mathcal{R}}}{\left[ \sum_{\mathbf{a}} \frac{(\bar{\pi}(\mathbf{a}|\mathbf{s}) - \bar{\pi}_\beta(\mathbf{a}|\mathbf{s}))^2}{\bar{\pi}_\beta(\mathbf{a}|\mathbf{s})} \right]}.$ \hfill (14)

The above derivation implies that each value iterate incurs some underestimation $\hat{V}^k(\mathbf{s}) \leq \hat{V}^{k-1}(\mathbf{s})$ when conservative hyper-parameter $\alpha$ satisfies some constraints. Now we compute the fixed point of value iteration (13) as the equivalency to $\hat{V}^k(\mathbf{s})$ when $k \to \infty$ and get the following estimated policy value $\hat{V}^{\bar{\pi}}(\mathbf{s})$ which shows a clear lower bound to actual state value function $V^{\bar{\pi}}(\mathbf{s})$, given that the simulation error $\left[ (I - \gamma T^{\bar{\pi}})^{-1} \mathbb{E}_{\bar{\pi}} \frac{\hat{\pi}_\beta}{\bar{\pi}_\beta} \right]$ can be avoided because of our off-policy setting:

$$
\hat{V}^{\bar{\pi}}(\mathbf{s}) = V^{\bar{\pi}}(\mathbf{s}) - \alpha \left[ \mathcal{E}_{\bar{\pi}} \left[ \frac{\bar{\pi}}{\bar{\pi}_\beta} - (1 + \frac{\tilde{\mathcal{R}}}{\alpha}) \right] \right]. \tag{15}
$$

Then, by further incorporating the sampling error $\left[ (I - \gamma T^{\bar{\pi}})^{-1} \frac{C_{r,T,\delta} R_{\max}}{1 - \gamma \sqrt{|D|}} \right](\mathbf{s})$, we get the following full evidential constraint for $\alpha$, which is a more strict constraint compared to CQL:

$$
\alpha \geq \frac{C_{r,T,\delta} R_{\max}}{1 - \gamma} \cdot \max_{\mathbf{s} \in D} \frac{1}{\sqrt{|D(\mathbf{s})|}} \left[ \sum_{\mathbf{a}} \frac{(\bar{\pi}(\mathbf{a}|\mathbf{s}) - \bar{\pi}_\beta(\mathbf{a}|\mathbf{s}))^2}{\bar{\pi}_\beta(\mathbf{a}|\mathbf{s})} \right]^{-1}.
$$

### C.2 Proof of Conservative Policy Improvement

**Lemma 3** (Conservative Policy Improvement). *Given an optimal policy $\pi_*$ that is the fixed point under action optimization using Equation (7), then the policy $\pi^*(\mathbf{a}|\mathbf{s})$ is a $\zeta$-safe policy improvement over behavior policy $\pi_\beta$ in the actual MDP $M$. The expected discounted return attained by a policy $\pi^*$ in the actual underlying MDP $M$, i.e., $J(\pi^*, M)$, is guaranteed to be higher than that attained by its behavior policy $\pi_\beta$ with a lowest $\zeta$ bound. Formally, we represent it as $J(\pi^*, M) \geq J(\pi_\beta, M) - \zeta$ with a high probability $1 - \delta$ where $\zeta$ is given as:*

$$
\begin{aligned}
\zeta = 2 \left( \frac{C_{r,\delta}}{1 - \gamma} + \frac{\gamma R_{max} C_{T,\delta}}{(1 - \gamma)^2} \right) \mathbb{E}_{\mathbf{s} \sim d_{\hat{M}}^\pi(\mathbf{s})} & \left[ \frac{\sqrt{|A|}}{\sqrt{|D(\mathbf{s})|}} \sqrt{D_{ECQL}(\bar{\pi}, \bar{\pi}_\beta)(\mathbf{s}) + 1} \right] \\
& - \alpha \frac{1}{1 - \gamma} \mathbb{E}_{\mathbf{s} \sim d_{\hat{M}}^{\pi^*}(\mathbf{s})} [D_{ECQL}(\pi^*, \bar{\pi}_\beta)(\mathbf{s})].
\end{aligned}
$$

*where $\delta$ is a high probability control related to $\zeta$, $d_{\hat{M}}^\pi$ is the state distribution from the empirical MDP $\hat{M}$. $|A|, |D|$ is the magnitude (norm) of action and state spaces. $C_{r,\delta}$ and $C_{T,\delta}$ are constants dependent on the concentration properties (variance) of evidential reward $r_\pi^e(\mathbf{a}, \mathbf{s})$ and state transition matrix $T(\mathbf{s}'|\mathbf{s}, \mathbf{a})$, respectively. $\gamma$ is the discounted factor. $D_{ECQL}$ is an evidential conservative factor ensuring the conservatism in policy evaluation.*

**Proof.** In this section, we first show that this policy improvement procedure defined in Equation (7) actually optimizes a penalized RL objective $J(\pi, \hat{M}) - \alpha \frac{1}{1-\gamma} \mathbb{E}_{\mathbf{s} \sim d_{\hat{M}}^{\pi}(\mathbf{s})}[D_{ECQL}(\bar{\pi}, \bar{\pi}_\beta)(\mathbf{s})]$ using Lemma D.3.1 following (Kumar et al., 2020), where $J(\pi, \hat{M})$ is the empirical discounted return of policy $\pi$ in empirical MDP $\hat{M}$, $D_{ECQL}$ is the conservatism term given in (14), and then we relate the performance of $\pi^*(\mathbf{a}|\mathbf{s})$ updated with this penalized RL objective, to the performance of itself and its behavior policy $\bar{\pi}_\beta$ in the actual MDP $M$ under the non-evidential settings so that we can leverage existing Theorem D.4 proposed by Kumar et al. (Kumar et al., 2020), thus gives:

$$J(\pi^*, M) \geq J(\bar{\pi}_\beta, M) - 2 \left( \frac{C_{r,\delta}}{1-\gamma} + \frac{\gamma R_{\max} C_{T,\delta}}{(1-\gamma)^2} \right) \mathbb{E}_{\mathbf{s} \sim d_{\hat{M}}^{\pi^*}(\mathbf{s})} \left[ \frac{\sqrt{|A|}}{\sqrt{|D|}} \sqrt{D_{ECQL}(\pi^*, \bar{\pi}_\beta)(\mathbf{s}) + 1} \right]$$
$$+ \alpha \frac{1}{1-\gamma} \mathbb{E}_{\mathbf{s} \sim d_{\hat{M}}^{\pi^*}(\mathbf{s})}[D_{ECQL}(\pi^*, \bar{\pi}_\beta)(\mathbf{s})]. \tag{16}$$

Note that all symbols here are in their non-evidential definitions which follow the same constraint on conservatism hyper-parameter $\alpha$ defined in (14). The explicitly expressed evidential term is contained in $D_{ECQL}(\pi^*, \bar{\pi}_\beta)(\mathbf{s})$. The proof for this statement is divided into two parts. The first part involves relating the return of $\pi^*$ in the empirical MDP $\hat{M}$ with the return of $\bar{\pi}_\beta$ in $\hat{M}$. Since, $\pi^*(\mathbf{a}|\mathbf{s})$ optimizes the penalized RL objective, it is the best policy under empirical MDP $\hat{M}$, and is guaranteed to behave better than the behavior policy $\bar{\pi}_\beta$ in a lowest bound governed by $\alpha \frac{1}{1-\gamma} \mathbb{E}_{\mathbf{s} \sim d_{\hat{M}}^{\pi^*}(\mathbf{s})}[D_{ECQL}(\pi^*, \bar{\pi}_\beta)(\mathbf{s})]$ as shown in the last term in Equation (16). The next step involves using concentration inequalities to upper and lower bound $J(\pi^*, \hat{M})$ and $J(\pi^*, M)$ and the corresponding difference for the behavior policy. According to Kumar et al.(Kumar et al., 2020), they apply Lemma D.4.1 to control such difference with $\gamma$, $R_{\max}$, and $C_{T,\delta}, C_{r,\delta}$.

### C.3 Proof of Theorem 2

We introduce the detailed derivative of Theorem 2 as given below:

$$J_{Q_c^e}(\theta) = \min_Q \alpha \left( \mathbb{E}_{\mathbf{s} \sim D, \mathbf{a} \sim \pi(\mathbf{a}|\mathbf{s})}[Q(\mathbf{a}, \mathbf{s})] - \mathbb{E}_{\mathbf{s} \sim D, \mathbf{a} \sim \pi_\beta(\cdot|\mathbf{s})}[Q(\mathbf{a}, \mathbf{s})] \right) + \frac{1}{2} \mathbb{E}_{\mathbf{s}, \mathbf{a}, \mathbf{s}' \sim D} \left[ \left( Q(\mathbf{a}, \mathbf{s}) - \hat{\mathcal{B}}^{\pi} \hat{Q}^k(\mathbf{a}, \mathbf{s}) \right)^2 \right]$$

$$= \min_Q \alpha \left( \mathbb{E}_{\mathbf{s} \sim D, \mathbf{a} \sim \pi(\mathbf{a}|\mathbf{s})}[Q(\mathbf{a}, \mathbf{s})] - \mathbb{E}_{\mathbf{s} \sim D, \mathbf{a} \sim \pi_\beta(\cdot|\mathbf{s})}[Q(\mathbf{a}, \mathbf{s})] \right)$$
$$+ \frac{1}{2} \mathbb{E}_{\mathbf{s}, \mathbf{a}, r_\pi^e, \mathbf{s}' \sim D, \mathbf{a}^* \sim \pi(\cdot|\mathbf{s}')} \left[ \left( Q(\mathbf{a}, \mathbf{s}) - \left( r_\pi^e(\mathbf{s}, \mathbf{a}) + \gamma \times \min\{\tilde{Q}^k(\mathbf{s}', \mathbf{a}^*), \hat{Q}^k(\mathbf{s}', \mathbf{a}^*)\} \right) \right)^2 \right]$$

$$= \min_Q \alpha \left( \mathbb{E}_{\mathbf{s} \sim D, \mathbf{a} \sim \pi(\mathbf{a}|\mathbf{s})}[Q(\mathbf{a}, \mathbf{s})] - \mathbb{E}_{\mathbf{s} \sim D, \mathbf{a} \sim \pi_\beta(\cdot|\mathbf{s})}[Q(\mathbf{a}, \mathbf{s})] \right) + \frac{1}{2} \mathbb{E}_{\mathbf{s}, \mathbf{a}, r_\pi^e, \mathbf{s}' \sim D, \mathbf{a}^* \sim \pi(\cdot|\mathbf{s}')}$$

$$\left( Q(\mathbf{a}, \mathbf{s}) - \left( \underbrace{\frac{1}{N} \sum_{i \in \mathcal{N}_u} \text{rating}_{u,i}}_{\mathbf{r}} + \lambda \underbrace{\frac{1}{N} \sum_{i \in \mathcal{N}_u} \mathcal{U}_\pi(\cdot|\mathbf{e}_i)}_{\mathbb{R}} + \gamma \times \min\{\tilde{Q}^k(\mathbf{s}', \mathbf{a}^*), \hat{Q}^k(\mathbf{s}', \mathbf{a}^*)\} \right) \right)^2$$

$$= \min_Q \alpha \left( \mathbb{E}_{\mathbf{s} \sim D, \mathbf{a} \sim \pi(\mathbf{a}|\mathbf{s})}[Q(\mathbf{a}, \mathbf{s})] - \mathbb{E}_{\mathbf{s} \sim D, \mathbf{a} \sim \pi_\beta(\cdot|\mathbf{s})}[Q(\mathbf{a}, \mathbf{s})] \right)$$
$$+ \frac{1}{2} \mathbb{E}_{\mathbf{s}, \mathbf{a}, r_\pi^e, \mathbf{s}' \sim D, \mathbf{a}^* \sim \pi(\cdot|\mathbf{s}')} \left[ \left( \left( Q(\mathbf{a}, \mathbf{s}) - \mathbf{r} - \gamma \times \min\{\tilde{Q}^k(\mathbf{s}', \mathbf{a}^*), \hat{Q}^k(\mathbf{s}', \mathbf{a}^*)\} \right) - \mathbb{R} \right)^2 \right].$$

We use $\hat{\mathcal{B}}^{\bar{\pi}} \hat{Q}^k(\mathbf{a}, \mathbf{s})$ to replace $\mathbf{r} + \gamma \times \min\{\tilde{Q}^k(\mathbf{s}', \mathbf{a}^*), \hat{Q}^k(\mathbf{s}', \mathbf{a}^*)\}$ and by some mathematical reshaping, we get:

$$J_{Q_c^e}(\theta) = \min_{\hat{Q}^k} \frac{1}{2} \mathbb{E}_{\mathbf{s}, \mathbf{a}, r, \mathbf{s}' \sim D} \left[ \left( \hat{Q}^k(\mathbf{a}, \mathbf{s}) - \hat{\mathcal{B}}^{\bar{\pi}} \hat{Q}^k(\mathbf{a}, \mathbf{s}) \right)^2 \right] + \alpha \mathbb{E}_{\mathbf{s} \sim D, \mathbf{a} \sim \bar{\pi}(\cdot|\mathbf{s})}[\hat{Q}^k(\mathbf{a}, \mathbf{s})]$$
$$- \alpha \left( \mathbb{E}_{\mathbf{s} \sim D, \mathbf{a} \sim \bar{\pi}_\beta(\cdot|\mathbf{s})}(1 + \mathbb{R})[\hat{Q}^k(\mathbf{a}, \mathbf{s})] \right) + \underbrace{(\mathbb{R}^2 + \mathbb{R} \cdot \hat{\mathcal{B}}^{\bar{\pi}} \hat{Q}^k(\mathbf{a}, \mathbf{s}))}_{\mathcal{C}}. \tag{17}$$

where $\mathcal{C}$ is a constant not relating to the Q-value iterate.

## D   EVIDENTIAL CONSERVATIVE Q-LEARNING ALGORITHM

We provided detail training procedure for the ECQL method in Algorithm 1.

---

**Algorithm 1** Evidential Conservative Q-Learning

---

**Require:** Hyperparameters: $\alpha, \lambda, \tau$, episode size $M$ and step size $T$
  1: Initialize SSE: $\omega$, action network: $\phi$, evidence network: $\psi$, and critic network: $\theta$ , initial state: $\mathbf{s}_0$ and initial user embedding: $\mathbf{u}_0$ with $W^l = 2N$ items in sliding window $W_0$ from interaction history $H_u$, and Item Pool $\mathcal{I}$
  2: **for** each epoch **do**
  3:     **for** each user as an episode **do**
  4:         **for** $t \in T$ **do**
  5:             Compute state: $\mathbf{s}_t$ with  (4).
  6:             Compute action: $\mathbf{a}_t \sim \pi_\theta(.|\mathbf{s}_t)$
  7:             Compute evidential score using  (2)
  8:             Recommend $top\text{-}N$ items based on computed evidential score to form a recommendation list $\mathcal{N}_u$.
  9:             Compute rewards based on recommendation list $\mathcal{N}_u$ utilizing (3).
 10:             Add $(\mathbf{s}_t, \mathbf{a}_t, r^e_\pi(\mathbf{s}_t, \mathbf{a}_t), \mathbf{s}_{t+1}, done)$ into replay buffer
 11:             Move sliding window and take $\frac{W^l}{2}$ newly interacted items from $H_u$ and replace the other $\frac{W^l}{2}$ items with the top-N items from the recommendation list $\mathcal{N}_u$.
 12:         **end for**
 13:         **if** episode index number reaches $M$ **then**
 14:             Sample batched data from replay buffer and forward into the networks.
 15:             **repeat**
 16:                 Update critic network with (6)
 17:                 Update action network with (7)
 18:                 Update SSE network with (5)
 19:                 Update evidence network with (8)
 20:             **until** converged
 21:         **end if**
 22:     **end for**
 23: **end for**

---

## E   DETAILED EXPERIMENTAL SETUP AND ADDITIONAL RESULTS

In this section, we provide additional details on the experiments, including datasets, setting, evaluation metrics, and comparison baselines. We also report more results, including comparison with additional baselines on two larger scale datasets, ablation study and qualitative analysis.

### E.1   DETAILED EXPERIMENTAL SETUP

**Description of datasets.** Besides the four dynamic recommendation datasets as introduced in the main paper, we include one additional dataset Movielens-10M to evaluate how ECQL performs on large-scale recommendation problems. Details of all the datasets are summarized below:

- **Movielens-1M**[1]: This dataset includes 1M ratings provided by 6,040 anonymous users on 3,900 distinct movies from 04/2000 to 02/2003.
- **Movielens-100K**[2]: This dataset contains 100,000 ratings from 943 users on 1,682 movies. Each user at least rated 20 movies from September 19, 1997 through April 22, 1998.
- **Netflix** (Bennett et al., 2007): This dataset has around 100 million interactions, 480,000 users, and nearly 18,000 movies rated between 1998 to 2005. We pre-processed the dataset and selected 6,042 users with user-item interactions from 01/2002 to 12/2005.

---

[1]https://grouplens.org/datasets/movielens/1M/
[2]https://grouplens.org/datasets/movielens/100k/

- **Yahoo! Music rating** (Dror et al., 2012): The dataset includes approximately 300,000 user-supplied ratings, and exactly 54,000 ratings for randomly selected songs. The ratings for randomly selected songs were collected between August 22, 2006 and September 7, 2006.
- **Movielens-10M** [3]: This dataset contains 10,000,054 ratings applied to 10,681 movies by 71,567 users, which was released in January 2009. All users selected had rated at least 20 movies.

## E.2 TRAINING AND TEST SETTING

For training, given a user interaction history $H_u$, we continuously capture most recent $N$ items after current time step into a sliding window $W_t$, which starts from the beginning of the interaction history. Then, we input the currently interacted $N$ items into the model along with the recommended $N$ items. These top-$N$ newly recommended items are selected from the item pool $\mathcal{I}$ by their predicted ranking score given by (2). Note that the ranking score masks out all non-interacted, previously interacted or recommended items to avoid lack of reward supervision or potential data leakage. Given above training setting, we further clarify how we conduct our testing. First, we split the user set into training and test users. Training users are used to train our ECQL model with their interaction history. In testing, for each test user, we assume no prior knowledge of interaction history and run ECQL on each user directly for different step size based on the minimum session length limit in each dataset. In each step, the user embedding is updated through the agent recommended items only. Then, we report the average recommendation performance with three metrics **Precision@N**, **nDCG@N** and **Test Return**. As the agent continues to collect more user interactions, the recommendation performance will continue to improve as shown in Figure 5.

### E.2.1 EXPERIMENTAL SETTING

We consider each user an episode for the RL setting and split users into 70% as training users and 30% as test users. For each user, we select the first $W^l = 10$ interacted items from history $H_u$ to represent an initial state $\mathbf{s}_0$. In the next state, we utilize previous state representation and concatenate with five item embeddings from the sliding window and the other five item embeddings from RL based recommendation to generate current state $\mathbf{s}_t$ by passing through the SSE module. Then, the action network generates mean and covariance for a Gaussian policy from which an action is sampled. This action is further passed to the evidence network, which utilizes the item embeddings of all user interacted items (Item Pool $\mathcal{I}$) to produce corresponding evidence for each item. We use the setting of classification, where explicit ground-truth ratings are used as class labels. With that evidence, we compute the evidential score by evaluating evidence-based rating and uncertainty to rank those items, which provides a list of top-$N$ ($N = 5$) final recommendations. We then evaluate the evidential reward based on their ground-truth ratings and information gain, functioning as the trade-off between exploitation and exploration. We set discounted factor $\gamma = 1$ and set $\tau = 3$ as a threshold to identify if an item is positive, *i.e.*, whether its ground-truth rating is larger than or equal to the threshold ($\text{rating}_{u,i} \geq \tau$). In testing, the agent may recommend items not interacted by the user. In such cases, we assign a neutral rating $\tau$ for those non-interacted items, where we set $\tau = 3$. For training, we conduct 5 RL epochs, each with full training of all training users (episodes). Each epoch is equipped with an annealing $\lambda$ ranging from 1 to 0.1 to adjust the emphasis from exploration to exploitation as the knowledge of training users increases. For testing, we conduct only one RL epoch containing all test users, and we use an annealing $\lambda$ ranging from 0.5 to 0.1 across variable step sizes in four different (minimum session length) data sets. We report the corresponding recommendation performance step-wise averaged over all test users. We implement the experiments based on the PyTorch framework with two A-100 GPUs.

### E.2.2 COMPARISON BASELINES

We compare with dynamic, sequential, deep learning, bandit and reinforcement learning based models. More specifically,

- **Dynamic models** include standard dynamic matrix factorization model timeSVD++ (Koren, 2009) as the time-evolving latent factorization model and collaborative Kalman filtering (CKF) (Gultekin & Paisley, 2014).

---

[3] https://grouplens.org/datasets/movielens/10m/

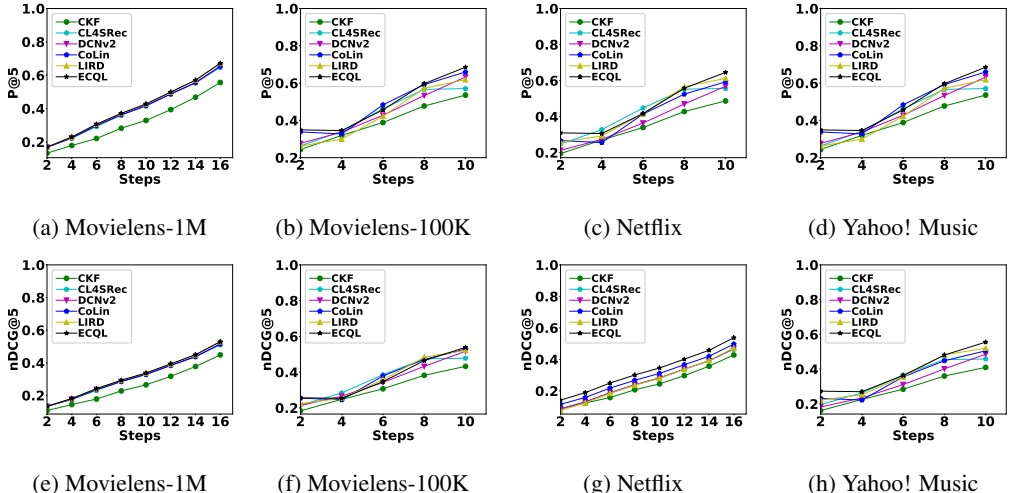

Figure 5: Performance comparison in each time step: (a)-(d): P@5; (e)-(h): nDCG@5

- **Sequential models** include Sequential Recommendation via Convolutional Sequence Embedding (Caser) (Tang & Wang, 2018), attention-based sequential recommendation model (SASRec) (Kang & McAuley, 2018), Adversarial Filtering Modeling(ADFM), Sampling-based Deep Interest Modeling (SDIM), Hierarchical Periodic Memory Network (HPMN) and sequential recommendation with bidirectional encoder (BERT4Rec) (Sun et al., 2019) for learning users' sequential preferences. For latest baselines, we have self supervised $S^3$-Rec (Zhou et al., 2020), contrastive learning based CL4SRec (Xie et al., 2022), and Seq2Seq (Ma et al., 2020) which reconstructs the representation of the future sequence as a whole and disentangle the intentions behind the given sequence of behavior.
- **Bandit models** include upper confidence bound (UCB-based) HLin (Wang et al., 2016), contextual bandit based method Lin (Li et al., 2010a) and CoLin (Wu et al., 2016) for news recommendation or collaborative environments respectively.
- **Reinforcement learning-based models** include $\epsilon$-greedy (Zhao et al., 2013), soft actor critic (SAC) (Haarnoja et al., 2018), deep Q-network based news recommendation (DRN) (Zheng et al., 2018), actor-critic based list-wise recommendation (LIRD) (Zhao et al., 2017), Smart Adaptive Recommendations (SAR) (Antaris & Rafailidis, 2021), Top-K Off-Policy Correction Recommendations (REINforCE) (Chen et al., 2019), Sequential Recommendation with Residual Actor for online exploration (ResAct) (Xue et al., 2022), Hyper-Actor Critic (HAC) (Liu et al., 2023), and Supervised Advantage Actor-Critic (SAS-SQN) (Silva et al., 2024).

### E.3 ADDITIONAL RESULTS

In this section, we present additional experiments and compare with different types of baselines.

**Step-wise recommendation performance.** We further show the step-wise performance of both precision@5 (P@5) and nDCG@5 metrics considering top-5 recommended items in all datasets as shown in Figure 5. We show the average precision and nDCG of all the test users over each step after the model is fully trained to demonstrate the effectiveness of the dynamic recommendation. We fixed the step size to 16, 10, 20, and 10 for the Movielens-1M, Movielens-100K, Netflix, and Yahoo! Music datasets based on their minimum number of user-item session length, respectively. At the initial steps, both precision and nDCG are low for all models (we choose the best model from each category as shown in Table 2). This is as expected due to lack of user interacted item observations. All the models start to improve after the initial stage. Dynamic models and sequential models still have poor performance compared to the RL-based methods. The proposed ECQL model provides consistently better performance over the entire process. However, it has a smaller advantage at the beginning due to its strong focus on exploration. After several step's observation, it quickly grasp user's interests and outperforms all its competitors by a clear margin.

**Statistical testing results.** We have conducted statistical tests by running our model along with most competitive baselines from each category three times and collecting the corresponding mean and standard deviation of P@5 and nDCG@5 performance on MovieLens-1M and MovieLens-100K datasets. The results are included in Table 7. It can be seen that the proposed model is more effective in performing effective exploration and provides more accurate recommendations, resulting in higher P@5 and nDCG@5 on both datasets considering the mean and variance in multiple runs. We further conduct a significance test to compare ECQL with the second best performing baseline LIRD. We obtain a p-value of 0.04, which confirms the performance advantage of ECQL over LIRD is statistically significant.

Table 7: Statistical testing results

| Model | MovieLens-1M | | MovieLens-100K | |
|---|---|---|---|---|
| | P@5 | nDCG@5 | P@5 | nDCG@5 |
| CL4SRec | 0.6135±0.019 | 0.4952±0.014 | 0.5813±0.018 | 0.4781±0.012 |
| CKF | 0.5567±0.008 | 0.4481±0.005 | 0.5285±0.012 | 0.4322±0.011 |
| DCNv2 | 0.6152±0.016 | 0.5187±0.013 | 0.6158±0.014 | 0.5166±0.012 |
| LIRD | 0.6238±0.025 | 0.5332±0.018 | 0.6137±0.023 | 0.5222±0.018 |
| CoLin | 0.6212±0.022 | 0.5236±0.016 | 0.6227±0.021 | 0.5295±0.016 |
| **ECQL** | **0.6313± 0.028** | **0.5365± 0.021** | **0.6379± 0.022** | **0.5386± 0.016** |

**Comparison with RL-based sequential methods on larger datasets.** In the main paper, we show that ECQL achieves a clear advantage over the baseline models on the four benchmark datasets. Now we focus on evaluating how ECQL performs on larger scale recommendation problems. We select the strongest baselines from each category for comparison with proposed ECQL on a subset selected from the large data set MovieLens-10M, denoted as ML-10M, with the minimum user interaction length larger than 300. From Table 8, it is clear that our method is taking advantage of systematic exploration and providing much better performance in larger datasets. In comparison to CoLin, LIRD has better performance due to its capability to leverage a complex reinforcement learning process to strengthen its user representation capability. However, it also fails to perform enough exploration and achieves lower performance as compared with our method.

Table 8: Comparison with recent RL-based sequential baselines on larger dataset ML-10M

| Model | ML-10M | |
|---|---|---|
| | P@5 | nDCG@5 |
| CL4SRec | 0.5138 | 0.4598 |
| CKF | 0.4952 | 0.4325 |
| DCNv2 | 0.5374 | 0.4762 |
| CoLin | 0.5574 | 0.4516 |
| LIRD | 0.5624 | 0.4642 |
| **ECQL** | **0.6425** | **0.5518** |

**Combining ECQL with other RL4Rec methods.** Our proposed ECQL could be generalized into any RL4Rec methods and helps to improve their recommendation performance. In Table 9, we use a recent RL4Rec sequential model *CL4SRec* as a test platform to show the performance improvement achieved by our ECQL comparing to other RL exploration strategies. We report the P@5 and nDCG@5 on a larger MovieLens-10M data set to show the results.

### E.4 ADDITIONAL ABLATION STUDY

**Impact of hyperparameter ($\lambda$).** The hyperparameter ($\lambda$) plays a critical role in recommending the top-$N$ items through generating corresponding ranking scores in test phase. We test three different settings: $\lambda = 0.1$, $\lambda = 0.5$, and gradually reducing $\lambda$ from 0.5 to 0.1. As can be seen from Figure 6, dynamically adjusting $\lambda$ achieves consistently better test returns on all datasets. This supports the intuition that in the early steps, a large $\lambda$ allows the model to conduct sufficient exploration. Once the model gains sufficient knowledge from the user preference and is able to make accurate

Table 9: Comparison between ECQL and other RL exploration strategies on another RL4Rec framework *CL4SRec*

| Model | MovieLens-10M | |
|-------|-------|-------|
| | P@5 | nDCG@5 |
| SAC | 0.5923 | 0.4917 |
| EQL | 0.6075 | 0.5028 |
| CoLin | 0.6159 | 0.5133 |
| $\epsilon$-greedy | 0.5138 | 0.4598 |
| ICM | 0.5568 | 0.4689 |
| RND | 0.5348 | 0.4496 |
| ECQL | 0.6315 | 0.5318 |

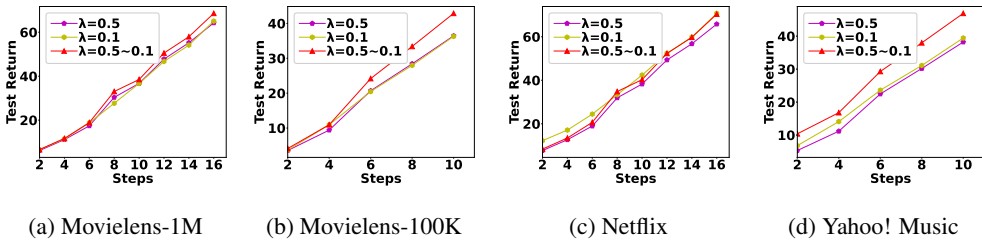

| (a) Movielens-1M | (b) Movielens-100K | (c) Netflix | (d) Yahoo! Music |
|---|---|---|---|

Figure 6: Test return of ECQL for different $\lambda$

recommendations, reducing $\lambda$ will allow the model to exploit its knowledge to provide effective recommendations.

**Impact of sequence encoders:** In this section, we investigate the impact of different sequence encoders, including RNN, LSTM, and GRU and report the result on Movielens-1M. Table 10 shows that our model's performance remains largely robust to different types of sequence encoders.

Table 10: Impact of sequential encoders

| Sequential encoder | P@5 | nDCG@5 |
|--------------------|------|--------|
| RNN + CEAC | 0.6313 | 0.5365 |
| LSTM + CEAC | 0.6312 | 0.5364 |
| GRU + CEAC | 0.6315 | 0.5366 |

**Impact of exploration strategies:** In the main paper, we show the test return from different exploration strategies to demonstrate the effectiveness of vacuity-guided exploration of ECQL. In this section, we further report the P@5 and nDCG@5 on MovieLens-1M to complement the test return result:

- Epsilon (*i.e.,* $\epsilon$-greedy) is a basic exploration strategy, which uses actor-critic to conduct Q-learning. Since we only set hyper-parameter $\epsilon = 0.1$, which is our best practice after grid-search, there's no need to combine conservative learning in this exploration direction.

- SAC also leverages actor-critic to conduct Q-learning. The difference comparing to $\epsilon$-greedy is that it uses first-order uncertainty "Entropy" in the rating prediction for score-based exploration and the reward design.

- EQL is our framework with traditional Q-learning in stead of conservative Q-learning, so the exploration is purely guided by uncertainty without any constraints.

- CoLin is a collaborative contextual bandit based method which explicitly models the underlying dependency among users/bandits with the help of a weighted adjacency graph, comparing to other bandit exploration technique like upper confidence bound based (UCB-based) HLin or factorUCB, it achieves best performance in multiple recommendation datasets.

Table 12: Recommended Movies for a random user

| Model | Recommended Important Movies | Movie Genre | Vacuity |
|---|---|---|---|
| ECQL | 1995,Star Trek: Voyager: Season 1 | **Sci-Fi,Action** | 0.10 |
| | 2005,7 Seconds | **Action,Adventure** | 0.12 |
| | 1994,Immortal Beloved | **Romance** | 0.14 |
| | 1996,No Way Back | **Action** | 0.11 |
| | 1996,Screamers | **Sci-fi, Action** | 0.09 |
| EQL | 2003,Dinosaur Planet | **Adventure**,Fantasy | 0.16 |
| | 1982,Nature: Antarctica | Documentary | 0.15 |
| | 1997,Sick | **Romance, Drama** | 0.14 |
| | 1994,Paula Abdul's Get Up | Educational | 0.16 |
| | 1997,Character | **Romance,Drama** | 0.15 |
| $\epsilon$-greedy | 1979,Winter Kills | **Action** | 0.08 |
| | 1991,Antarctica: IMAX | Documentary | 0.16 |
| | 1951,The Frogmen | **Drama** | 0.13 |
| | 1983,Silkwood | Documentary | 0.15 |
| | 2002,The Powerpuff Girls Movie | Animation | 0.15 |
| SAC | 1951,The Lemon Drop Kid | Comedy | 0.13 |
| | 2002,Mostly Martha | **Romance** | 0.13 |
| | 1989,A Fishy Story | **Romance** | 0.14 |
| | 2004,Spartan | **Action**,Military | 0.11 |
| | 1997,The Game | **Action** | 0.09 |
| CoLin | 1965,The Great Race | **Adventure,Action** | N/A |
| | 2002,Obsessed | **Romance,Drama** | N/A |
| | 2000,Magnolia: Bonus Material | Military | N/A |
| | 1965,The Battle of Algiers: Bonus Material | Military | N/A |
| | 1972,Seeta Aur Geeta | Musical,**Romance** | N/A |

- ICM uses curiosity to facilitate exploration, measured by the error in predicting the next state from the current state-action pair, leveraging a forward dynamic model trained on prior interactions. Additionally, it employs an inverse dynamic model to filter out irrelevant state representations.

- RND employs a fixed, randomly initialized network (random distillation network) as a reference model. A predictor neural network is trained to replicate the outputs of the reference model, and the prediction error for any new state serves as a novelty score.

Table 11: Impact of exploration strategies

| Model | P@5 | nDCG@5 |
|---|---|---|
| Epsilon | 0.5977 | 0.4834 |
| SAC | 0.6105 | 0.5215 |
| EQL | 0.6234 | 0.5331 |
| CoLin | 0.6212 | 0.5236 |
| ICM | 0.6057 | 0.5088 |
| RND | 0.6012 | 0.5024 |
| ECQL | 0.6313 | 0.5365 |

### E.5 ADDITIONAL QUALITATIVE ANALYSIS

We conduct an additional qualitative analysis of a particular user in the Netflix dataset by comparing the top-$N$ recommended items in time step 16 of our proposed method ECQL with other models using different exploration strategies, *e.g.,* $\epsilon$-greedy, SAC, and CoLin. We also include EQL to further validate the necessity of conservative Q-learning. As shown in Table 12, a user with ID 254775 has five most frequently watched movie categories. By order, they are **Action**, **Sci-Fi**, **Adventure**, **Drama**, and **Romance**, which form a top-5 frequency list and are bold in the table. From the table,

Table 13: Training Efficiency Comparison (averaged training convergence time in hours)

| Category | Model | GFLOPS | MovieLens-1M | MovieLens-100K |
|----------|-------|--------|--------------|----------------|
| Sequential | CL4SRec | 5.48 | 12.9 | 4.5 |
| Dynamic | CKF | 5.15 | 11.0 | 3.8 |
| Deep Learning | DCNv2 | 5.52 | 13.3 | 4.6 |
| Bandit | CoLin | 5.85 | 13.9 | 5.0 |
| Reinforce | LIRD | 5.87 | 14.0 | 5.0 |
| Proposed | **ECQL** | 5.08 | 12.0 | 4.2 |

we observe that ECQL successfully captures user's long-term interests by recommending important movies that all belong to the top-5 frequency list. Compared to ECQL, EQL recommends some irrelevant movies of Documentary and Educational Genres that deteriorate the performance since it is not constrained by a conservative view in the Q-learning process. Frequently recommending totally irrelevant movies may pose a higher risk of losing the user. For different exploration strategies, $\epsilon$-greedy brings the least performance by recommending only two important movies that lie in the top-5 frequency list as it does not leverage a systematic exploration way to explore the user's potential interests. In comparison, bandit-based CoLin and entropy-based SAC leverage bandit theory and first-order entropy to measure the uncertainty and both lead to an improved recommendation performance by recommending at least three important high frequent movies out of five that reflect the user's interests. However, we emphasize that they only capture the short-time interest of the user, as demonstrated by their recommended important items' relative order in the top-5 frequency list.

The above analysis is from a random test user in the Netflix data set. For a more comprehensive analysis, we calculate the Gini index of the diversity for all the recommended results. First, we test 1,800 users from Netflix test set and then collect 16 (number of time steps) $\times$ 1,800 total recommendations. For each recommendation, we calculate its Gini index by separating 5 results into different categories and then calculate

$$\text{Gini} = 1 - \sum_{c=1}^{C} P(c)^2 \tag{18}$$

where $C$ is the number of categories and $P(c), c \in [1, C]$ is the probability for each category. Finally, we average the calculated Gini index for 1,800 recommendations and get the averaged Gini index as 0.71, which is close to 1. This indicates that the recommendation of our model is quite diverse and contains objects from different categories. Comparing to SAC and CoLin which have averaged Gini indexes 0.65 and 0.68 respectively, our ECQL achieves the most diverse recommendation thanks to the novel evidence-based exploration, which is also verified by Table 12.

### E.6 TRAINING EFFICIENCY

We report the training efficiency (averaged training convergence time measured in hours) comparison of the ECQL model on MovieLens-100K and MovieLens-1M datasets with respective strongest baseline in each category in Table 13. For fair comparison, in all settings, the training resource is two A-100 GPUs of memory 80G and all baselines are implemented with PyTorch framework. Besides, our model's computation per top-5 recommendation step reaches an average 5.08 GFLOPS.

## F LIMITATIONS, FUTURE WORKS, AND BROADER IMPACT

In our evaluation, we applied the proposed ECQL method to the simulated off-policy setting using the offline user interaction datasets to mimic real-world scenarios. It could be more interesting to implement this ECQL method in the online scenarios which could better reflect the model's real-time performance. We are planning to further extend this work to apply to those real-world time sequence data in different fields such as health and autonomous vehicle to see its scalability, generalizability, and adaptability as a future work. And we will also conduct plenty of ablation and case analysis in these sparse and safety-critical domains to evaluate the effectiveness of our unique evidence-based exploration balanced with a conservative learning design.

The proposed ECQL can be generally applied to safety-critical applications, which are common in many domains such as health, autonomous vehicle, cybersecurity, military operations, and more. We provide conservative evidential exploration which doesn't let learned policy deviate far from the behavioral policy and supports the gradual exploration of a complex environment with sparse reward signals by leveraging fine-grained second-order uncertainties. The principled exploration can ensure high information gain with much reduced data annotations, which can benefit many domains where data annotation is costly.

## G   SOURCE CODE

The source code and processed datasets can be accessed here. `https://github.com/ritmininglab/ECQL`

