# OpenReview forum: "Looking into User’s Long-term Interests through the Lens of Conservative Evidential Learning"
_ICLR.cc/2025/Conference — ICLR 2025 Poster_

### Official Review · Reviewer_dEvt · 2024-10-21

**Soundness:** 3
**Presentation:** 3
**Contribution:** 3
**Rating:** 8
**Confidence:** 4

**Summary:**

This paper proposes a model based on evidential conservative Q-learning for sequential recommendation. It integrates the evidential uncertainty into the rating scores to encourage exploration in reinforcement learning. Meanwhile, conservative regularization terms are added to avoid the over-optimistic estimation of the Q-values. Some theoretical analyses are given to justify proposed methods. Experiments on several datasets empirically validate the superior performance with compared baselines on certain metics.

**Strengths:**

1.	The studied problem is important. Building a model to pursue long-term reward while making efficient exploration and avoiding user dissatisfaction is a valuable and challenging topic for both academic and industry.
2.	Modeling the uncertainty by introducing vacuity of evidence is interesting and novel.
3.	Theoretical proofs are given to justify proposed methods, building connections between the evidential uncertainty and the conservative policy update.
4.	Empirical experimental results are promising, revealing superior performance over several baselines.

**Weaknesses:**

1.	Based on the definition in Eq. (2), it appears that the candidate item pool throughout the training iterations is restricted to items with which the user has interacted. This limitation may introduce inconsistencies between training and inference and lead to bias issues.
2.	In Eq. (2), it says that 'Wt indexes the time step reached by the current sliding window.' I’m unclear about the definition of 'time step' and its correlation with the original appearance position of a recommended item. Furthermore, it appears that this down-weight term is absent from both their theoretical and experimental analyses, raising questions about its necessity and rationale.
3.	It says that “During testing, for item i′ not appearing in user u’s interaction history Hu, a neutral rating ratingu, i′ = τ will be assigned to give neutral feedback”.  This will lead to complete overlook of all missing user-item interaction values, which can make the evaluation results highly biased on metrics except for NCIS.
4.	It's somewhat puzzling that a model optimized for long-term rewards can outperform state-of-the-art baselines, which are optimized for immediate rewards, on one-step metrics such as Precision and NDCG.

**Questions:**

All my questions are listed in the weakness part.

---

> ### Author Response · Authors · 2024-11-23
>
> Thank you for providing constructive comments and suggestions. We provide our detailed response below.
>
> ---
> **Q1: Based on the definition in Eq. (2), it appears that the candidate item pool throughout the training iterations is restricted to items with which the user has interacted. This limitation may introduce inconsistencies between training and inference and lead to bias issues.**
>
> Thank you for this insightful comment. We agree that during the training phase, only user-interacted items are used to train the RL agent, as they provide real ground-truth ratings. For non-interacted items that rank high in predictions, the agent masks them out and selects only interacted items for reward evaluation. In the testing phase, we assign a neutral feedback score (3 on a 1–5 scale) to such recommended non-interacted items to ensure an unbiased evaluation. The comparison results remain fair, as all baselines are evaluated under the same training and testing conditions. Additionally, the sparse reward space during training encourages our model to explore novel items—those on which it currently lacks knowledge. These items often signify critical shifts in user interests and present opportunities for potentially higher rewards in the long term.
>
>
> **Q2: In Eq. (2), it says that 'Wt indexes the time step reached by the current sliding window.' I’m unclear about the definition of 'time step' and its correlation with the original appearance position of a recommended item. Furthermore, it appears that this down-weight term is absent from both their theoretical and experimental analyses, raising questions about its necessity and rationale.**
>
> Thank you for the question. User interaction history, $H_u$, is considered as a sequence of interacted items ordered by time, with $W_t$ representing an observation window sliding over it. The "time step" reached by the sliding window at time $t$ corresponds to the position of the last item's appearance within the window, preceding any future recommended item's position. In the revised paper, we refer to this as "the last experienced item's appearance position." The down-weight term is designed to prevent distant items from being recommended during training. It does not affect our theoretical analysis and has minimal impact on our empirical results, which is why we did not include further analysis.
>
> **Q3: It says that “During testing, for item $i'$ not appearing in user $u$’s interaction history $H_u$, a neutral rating rating, $i' = \tau$ will be assigned to give neutral feedback”. This will lead to complete overlook of all missing user-item interaction values, which can make the evaluation results highly biased on metrics except for NCIS.**
>
> We clarify that this approach is reasonable given the notoriously sparse reward in recommender systems (RS) and the lack of actual feedback for many items. While the model disregards missing interaction values during training, the test comparison results remain unbiased because:
>
> 1. Neutral feedback is assigned to non-interacted items, ensuring fairness in evaluating user preferences.
>
> 2. All baselines are assessed under the same training and testing conditions.
>
> To further validate the superior performance of ECQL, we provide a comparison of RL baseline performance using NCIS in Table 3 of the main paper. NCIS, a variant of average cumulative reward, is weighted by the normalized capped importance sampling score derived from policy probabilities.
>
> **Q4: It's somewhat puzzling that a model optimized for long-term rewards can outperform state-of-the-art baselines, which are optimized for immediate rewards, on one-step metrics such as Precision and NDCG.**
>
> Thank you for this insightful comment. We agree with the reviewer that nDCG@N and Precision@N are primarily designed to measure one-step recommendation performance. To place one-step performance in the context of the dynamic recommendation process, we include Figure 5 in the Appendix, which shows that both nDCG@N and Precision@N improve as the step count increases. This result highlights that by accumulating experience from the user's interaction history, the model gradually refines its state representation, effectively capturing the user's long-term preferences. This improvement is directly reflected in the higher recommendation quality in later steps. We also clarify that these metrics are widely used in related works, along with other one-step metrics like HR@N [1].
>
>
> ---
>
> **References**
>
> [1] Antaris, Stefanos, and Dimitrios Rafailidis. "Sequence adaptation via reinforcement learning in recommender systems." Proceedings of the 15th ACM Conference on Recommender Systems. 2021.

---

> > ### Author Response · Authors · 2024-11-29
> > **Follow-up from the authors and reminder of the reviewer-author discussion deadline [December 2nd]**
> >
> > Dear Reviewer dEvt,
> >
> > Thank you for your excellent questions. In our rebuttal, we have clarified the definition of "time step" in Equation 2 and made revisions to the paper accordingly. Additionally, we have justified the fairness of our experimental setup, addressing the following points:
> >
> > - The fairness of only levering user interacted items in training as they provide real ground-truth ratings and assigning neutral feedback for recommended non-interacted items in testing for an unbiased evaluation. We clarify that all baselines are evaluated under the same training and testing condition for fair evaluation.
> >
> > - The use of one step metrics such as Precision and NDCG to measure long-term reward. By accumulating experience from the user’s interaction history, the model gradually refines its state representation, effectively capturing the user’s long-term preferences, reflected in the higher recommendation quality in later steps.
> >
> > We deeply appreciate the reviewer’s careful reading of our paper and the insightful questions raised. Addressing these points has allowed us to better justify our methodology and experimental setup. As always, your feedback has significantly strengthened our work, and we are happy to address any additional questions you may have.
> >
> > Sincerely,
> >
> > Authors

---

> > > ### Comment · Reviewer_dEvt · 2024-11-30
> > >
> > > Thanks for the detailed response and illustration! I'll raise the score to 8.

---

> > > > ### Author Response · Authors · 2024-11-30
> > > >
> > > > We sincerely thank the reviewer for taking the time to read our response and for your thoughtful feedback. We greatly appreciate your support of our work!

---

### Official Review · Reviewer_bGtM · 2024-11-02

**Soundness:** 3
**Presentation:** 3
**Contribution:** 3
**Rating:** 6
**Confidence:** 2

**Summary:**

This paper proposes a framework called Conservative Evidential Q-Learning (ECQL), which aims to learn effective and robust recommendation strategies by combining evidential uncertainty with conservative learning. ECQL leverages evidence-based exploration to discover items that may lie beyond the current observation range but reflect users' long-term interests. By evaluating strategies from an uncertainty-driven conservative perspective, ECQL reduces the risk of recommending items that deviate excessively from users' current interests.The core components include a Sequential State Encoder, which combines users' historical information with a sliding window that encompasses both current interactions and new items recommended through reinforcement learning exploration, and a Conservative Evidential Actor-Critic Module, which maximizes conservative evidential Q-values to predict scores based on evidence and explores the item space using an uncertainty-based ranking score.Experimental results demonstrate that ECQL excels at capturing users' long-term interests, progressively recommending items that are not significantly different from users' current preferences. This maintains recommendation relevance even with sparse interactions.

**Strengths:**

This paper provides a highly detailed and well-structured presentation of its content. The problem statement is thoroughly explained, supported by experimental results that clearly validate the proposed framework's effectiveness. Each experimental result is precise and comprehensive. The theoretical derivations are carefully detailed, making it easier for readers to follow the logical flow and understand the underlying principles. In addition, the extensive experimentation reinforces the paper’s claims, offering solid evidence for the model’s advantages in capturing long-term user interests and providing stable recommendations. Overall, the combination of clear theoretical foundations with rigorous experimental validation makes this paper a valuable contribution to the field.

**Weaknesses:**

1. The abstract does not mention the motivation behind the proposed method, which might leave readers somewhat confused.

2. Introducing uncertainty into exploration strategies in reinforcement learning has already been extensively studied, so the novelty of this paper is not particularly outstanding.

3. The paper could benefit from adding some related work on the use of uncertainty in recommendation systems.

4. It would be helpful to experimentally verify whether the evidence network’s evidential score can serve as a plug-and-play component in an ε-greedy strategy to improve the performance of other RL4Rec methods.

**Questions:**

1. The motivation for introducing an uncertainty-aware exploration strategy is not clearly explained. The correspondence between the solution and the problem is unclear, and it doesn’t explain why uncertainty-aware exploration can effectively capture user’s evolving preferences and achieve the maximum expected reward over the long term, while existing methods cannot.

2. There is generally a balance between exploration diversity and accuracy performance. Further explanation is needed on how the proposed uncertainty-based exploration strategy reflects this balance.

---

> ### Author Response · Authors · 2024-11-23
>
> Thank you for providing constructive comments and suggestions. We provide our detailed response below.
>
> ---
> **Q1: The abstract does not mention the motivation behind the proposed method. The motivation for introducing an uncertainty-aware exploration strategy is not clearly explained.**
>
> Thank you for the suggestion. We have updated the abstract to clarify the motivation of our work. As further elaborated in the introduction, most existing methods treat recommendation as a static process, which prevents them from effectively accounting for users’ evolving preferences. Sequential recommendation methods address this limitation to some extent by leveraging previously interacted items to capture users’ dynamic behavior. However, prior RL-based recommender system models largely rely on standard exploration strategies, such as $\epsilon$-greedy, which are less effective in scenarios with a large item space and sparse reward signals due to limited user interactions. As a result, these methods may struggle to learn an optimal policy that adequately captures users’ evolving preferences and achieves the maximum expected reward over the long term. The qualitative results presented in Figure 1 and Table 1 illustrate the limitations of existing approaches and further highlight the need for a systematic, uncertainty-aware exploration strategy.
>
>
>
>
> **Q2: Introducing uncertainty into exploration strategies in reinforcement learning has already been extensively studied, so the novelty of this paper is not particularly outstanding.**
>
> Please refer to MR2 in  our general response for a detailed clarification on the novelty of the proposed method.
>
>
> **Q3: It would be helpful to experimentally verify whether the evidence network’s evidential score can serve as a plug-and-play component in an $\epsilon$-greedy strategy to improve the performance of other RL4Rec methods.**
>
> Thank you for this excellent suggestion. Indeed, our proposed ECQL framework can be generalized to any RL4Rec method. To demonstrate this, we use the recent RL4Rec model **CL4SRec** as a test platform to evaluate the performance improvements achieved by incorporating ECQL compared to other RL exploration strategies. The results, presented in the table below, report P@5 and nDCG@5 metrics on the larger MovieLens-10M dataset.
>
> | | P@5 | nDCG@5 |
> |-----------------|------------|-----------------|
> | SAC | 0.5923 | 0.4917 |
> | EQL | 0.6075 | 0.5028 |
> | CoLin |  0.6159 | 0.5133 |
> | $\epsilon$-greedy |  0.5138 | 0.4598  |
> | ICM | 0.5568 | 0.4689 |
> | RND | 0.5348 | 0.4496 |
> | ECQL |  0.6315 | 0.5318 |
>
> **Q4: The paper could benefit from adding some related work on the use of uncertainty in recommendation systems.**
>
> Thank you for the suggestion. We have revised our additional related work section in Appendix B to include additional related work on using uncertainty in recommendation systems.
>
> **Q5: There is generally a balance between exploration diversity and accuracy performance. Further explanation is needed on how the proposed uncertainty-based exploration strategy reflects this balance.**
>
> The balance between exploration diversity and accuracy performance is achieved by the parameter $\lambda$ in the evidential reward as defined in Eq. (3). We study the impact of $\lambda$ and report the results in Figure 6 of Appendix E.4. We test three different settings: $\lambda=0.1$, $\lambda=0.5$, and gradually reducing $\lambda$ from $0.5$ to $0.1$.
> As can be seen from Figure 6, dynamically adjusting $\lambda$ achieves consistently better test returns on all datasets.
> This supports the intuition that in the early steps, a large $\lambda$ allows the model to conduct sufficient exploration. Once the model gains sufficient knowledge from the user preference and is able to make accurate recommendations, reducing $\lambda$ will allow the model to exploit its knowledge to provide effective recommendations.

---

> > ### Author Response · Authors · 2024-11-29
> > **Follow-up from the authors and reminder of the reviewer-author discussion deadline [December 2nd]**
> >
> > Dear Reviewer bGtM,
> >
> > Thank you once again for your thoughtful comments and questions! In our rebuttal, we have clarified the motivation and novelty of our work by emphasizing the limitations of existing approaches. This underscores the necessity of an RL framework designed to optimize long-term rewards through a conservative and evidential uncertainty-aware exploration strategy. Furthermore, we demonstrate the versatility of our model as a plug-and-play component by benchmarking it against other RL exploration strategies using CL4SRec as a test platform.
> >
> > We believe that addressing your suggestions has significantly strengthened our paper, and we greatly appreciate your support. We hope our responses meet your expectations, and we remain more than happy to address any additional questions or concerns you may have.
> >
> > Sincerely,
> >
> > Authors

---

### Official Review · Reviewer_JFxC · 2024-11-03

**Soundness:** 2
**Presentation:** 2
**Contribution:** 2
**Rating:** 3
**Confidence:** 4

**Summary:**

This paper presents a approach to recommenders which aims to capture users' long-term interests through reinforcement learning and evidential uncertainty. The authors propose ECQL to address limitations in existing recommender systems, particularly in capturing evolving user preferences and long-term interests. The framework integrates evidence-based uncertainty and conservative learning to develop a conservative recommendation policy.
ECQL employs a typical sequential state encoder that generates the current state of the environment by aggregating,
A sliding window containing current user interactions, older actions and a newly recommended items from RL exploration
This approach allows the model to represent both short-term and potential long-term user interests.
Another module seems to perform rating prediction by maximizing the conservative evidential Q-value

**Strengths:**

The experimental results seem to somewhat validate the approach

**Weaknesses:**

The paper is extremely dense written and difficult to follow, moreover the approach is over-engineered (see figure) . It is unclear what and by how much each component is contributing to the model. It is unclear if all these components are needed. Moreover the datasets used to not seem to have strong sequential behavioural patterns and the reliance on ratings makes the method somewhat irrelevant to modern recommender systems.

**Questions:**

I suggest the authors try and simplify both the method and the presentation of the work.

---

> ### Author Response · Authors · 2024-11-23
> **Response to Reviewer JFxC [Part I]**
>
> Thank you for providing constructive comments and suggestions. We provide our detailed response below.
>
> ---
> **Q1: The paper is dense and difficult to follow, the approach is over-engineered.**
>
> As discussed in the introduction section of the paper, the proposed ECQL framework is *designed to address the key limitations of existing reinforcement learning (RL) based recommendation systems*. These methods primarily rely on standard exploration strategies (e.g., $\epsilon$-greedy), making them less effective to capture user's long-term interest. To fill out this critical gap, we conduct novel evidential conservative Q-learning (ECQL) that utilizes a balanced exploitation (with high predicted ratings) and exploration (with evidential uncertainty) strategy for effective recommendations. We formulate an evidential RL framework that augments the reward based RL objective with evidential uncertainty to facilitate the exploration of unknown items. The **evidential uncertainty** formulation substantially improves exploration and robustness by acquiring diverse behaviors that are indicative of a user's long-term interest. Additionally, we encourage the model to explore items that do not significantly deviate from users' current interests given the sparse interactions. Such gradual exploration is guided by a **conservative evidential Q-value** that prevents recommending totally irrelevant items, causing user frustrations.
>
> The key technical objectives as summarized above are achieved through two major components: a **sequential state encoder** and a **Conservative Evidential Actor-Critic (CEAC) module**. The former primarily focuses on generating the current state of the environment by aggregating the previous state, the current items captured by a sliding window, and the future items from the recommendation. It provides an effective means of dynamic state representation for better future recommendations. The CEAC module leverages evidential uncertainty to effectively explore the item space to recommend items that potentially align with the user's long-term interest. It encourages learning the optimal policy by maximizing a novel conservative evidential Q-value to make more diverse recommendations that may reflect a long-term interest while keeping a conservative view that does not deviate too much from current interests.
>
> As a RL model, ECQL includes all the essential RL components like other existing RL baselines, such as SAC [1] or HAC [2]. It mainly replaces the standard reward and Q-value with the novel evidential reward and conservative evidential Q-value without added complexity as compared with other RL baselines. **It achieves a training efficiency comparable to other RL baselines as shown in Table 12 of Appendix E.6.**
>
> **Q2: It is unclear how much each component is contributing to the model.**
>
> We have conducted a comprehensive ablation study to thoroughly investigate the contribution of each key component of ECQL. The results are presented in Section 5.3 and Appendix E.4. In particular,
>
> - In Figure 3, we investigate the effect of exploration of ECQL by comparing with two alternative designs: (1) without vacuity and (2) without conservative learning guided exploration. We also compare exploration using the first-order uncertainty, employed by soft-actor-critic (SAC).
>
> - In Table 5, we evaluate the impact of the two key modules of ECQL: SSE and CEAC. Results show that the integration of the two modules significantly outperforms each individual module.
>
> - In Appendix E.4, we further evaluate the impact of the exploitation-exploration balancing hyper-parameter $\lambda$, the sequence encoder, and exploration strategies. The results are presented in Figure 6, Table 10, and Table 11.
>
>
> **Q3: Datasets used seem not to have strong sequential behavioral patterns.**
>
> We would like to clarify that, broadly speaking, sequential data refers to data where the current instance depends on previous instances over time. For example, the Movielens dataset includes user interaction sequences with timestamps, where a user's current movie interaction depends on their past interactions. In this sense, all the datasets used in our evaluation are sequential, as they record how user interactions with items change over time. Moreover, standard sequential recommendation models such as CASER (Tang et al., 2018), SASRec (Kang et al., 2018), and BERT4Rec (Sun et al., 2019) have used these datasets for evaluation. Thus, we selected these datasets to ensure fair comparisons with established baselines. The strong recommendation performance suggests that significant sequential behavioral patterns are present in these datasets and have been effectively captured by the model.
>
> **Response continues...**

---

> ### Author Response · Authors · 2024-11-23
> **Response to Reviewer JFxC [Part II]**
>
> **Q4: Reliance on ratings is outdated for modern recommender systems.**
>
> We would like to clarify that we follow the standard process of applying reinforcement learning (RL) methods in recommender systems, where ratings are used to compute rewards, as demonstrated in recent baselines such as ResAct (Xue et al., 2023) and SAR (Antaris et al., 2021). Additionally, our evidential reward definition is general and can be extended to accommodate other types of user feedback beyond ratings. For instance, in the case of click-based feedback, ratings can be replaced with click frequency or rate, while the vacuity term (one key novelty of our approach) in Equation 3 remains unchanged.
>
> ---
> **References**
>
> [1] Hong, Kimura, et al. "A Soft Actor-Critic Algorithm for Sequential Recommendation." In Database and Expert Systems Applications: 35th International Conference, DEXA 2024, Naples, Italy, August 26–28, 2024.
>
> [2] Liu, Shuchang, et al. "Exploration and regularization of the latent action space in recommendation." Proceedings of the ACM Web Conference 2023. 2023.

---

> > ### Author Response · Authors · 2024-11-29
> > **Follow-up from the authors and reminder of the reviewer-author discussion deadline [December 2nd]**
> >
> > Dear Reviewer JFxC,
> >
> > Thank you once again for your thoughtful comments and questions! In this rebuttal, we clarify that ECQL, as an RL model, includes only the essential RL components found in existing baselines such as SAC and HAC. The novelty lies in the introduction of the evidential reward and conservative evidential Q-value mechanisms, which enhance performance without adding complexity compared to other RL baselines. We also reference multiple ablation studies in the paper and appendix, which demonstrate the contribution of each component to the model's overall performance. Additionally, regarding the dataset and reliance on ratings, we follow standard practices for applying reinforcement learning to recommender systems. We have clarified our experimental settings and provided comparisons to recent baselines, including ResAct (Xue et al., 2023) and SAR (Antaris et al., 2021).
> >
> > We hope you find our response satisfactory and would consider raising the score. As always, we are happy to address any additional questions or concerns you may have.
> >
> > Sincerely,
> >
> > Authors

---

### Official Review · Reviewer_mvAf · 2024-11-04

**Soundness:** 3
**Presentation:** 3
**Contribution:** 3
**Rating:** 8
**Confidence:** 5

**Summary:**

This paper aims to tackle the exploration problem for RL4RS algorithms. The paper propose a evidential conservative Q-learning framework (ECQL) that models the uncertainty of samples by an evidential network. The paper also controls the degree of exploration by a conservative critic update. Experiments on 4 datasets validates that ECQL outperforms dynamic models, sequential models, deep-learning models, bandit models, and stoa RL-based models.

**Strengths:**

Originality: The paper studies the exploration problem for RL-based RS, which is novel. Also, the application of the evidential network to quantify the exploration bonus is novel.
Quality: The paper does detailed and sufficient experiments in both recommendation metrics and rl-based metrics such as NCIS. Recent RL-based methods are compared. The paper also provides case studies on ECQL and SAC, about the relevance and the exploration ability.
Clarity: The paper is well written and easy to follow.
Significance: The paper proposes a new algorithm to tackle the exploration problem for RL-based RS. I like the idea of controlling exploration by a conservative Q-learning updating mechanism.

**Weaknesses:**

1.The paper does not open-source their code.
2.The paper does not discuss or compare other exploration methods that are widely used in RL, such as ICM and RND.
3.Why do online RL methods perform well in the offline learning setting, such as SAC,HAC in Table 2?

**Questions:**

See the weaknesses.

---

> ### Author Response · Authors · 2024-11-23
>
> Thank you for providing constructive comments and suggestions. We provide our detailed response below.
>
> ---
> **Q1: The paper does not open-source their code.**
>
>
> We have included the link to the source code in Appendix G.
>
>
> **Q2: The paper does not discuss or compare other exploration methods that are widely used in RL, such as ICM and RND.**
>
>
> Thank you for pointing out these relevant exploration methods! We have incorporated these works into our updated related work section and discussed their differences from our approach.
>
> ICM [1] and RND [2] are designed for complex deep reinforcement learning (DRL) environments (e.g., Atari games) with sparse reward signals, where exploring novel states is critical to uncover rare rewards, penalizing events, or turning points. ICM [1] uses curiosity to facilitate exploration, measured by the error in predicting the next state from the current state-action pair, leveraging a forward dynamic model trained on prior interactions. Additionally, it employs an inverse dynamic model to filter out irrelevant state representations. Similarly, RND [2] employs a fixed, randomly initialized network (random distillation network) as a reference model. A predictor neural network is trained to replicate the outputs of the reference model, and the prediction error for any new state serves as a novelty score.
>
> We adapted these exploration methods to our recommender system setting by incorporating inverse dynamic forward models and fixed randomly initialized networks. The results reported on MovieLens-1M are shown in the table below. As can be seen, ICM and RND perform worse than exploration baselines commonly used in recommender systems, such as SAC and CoLin, which in turn are outperformed by the proposed ECQL model. This performance gap arises because SAC, CoLin, and ECQL directly model rating prediction uncertainty to encourage novel item discovery, which is more effective than novel state discovery designed for high-dimensional DRL environments like Atari games.
>
> ECQL further outperforms other baselines due to its state representation tailored for recommender systems. ECQL systematically integrates past experiences, current interactions, and long-term recommendations using an RNN module. Additionally, its vacuity-guided exploration strategy prioritizes recommending items with the least known information (high vacuity). By collecting user feedback on such items, the RL agent gains valuable insights into user preferences, leading to improved recommendations in the long run.
>
>
> | | P@5 | nDCG@5 | R@16 |
> |-----------------|------------|-----------------|------------|
> | SAC | 0.6105  | 0.5215 | 61.63 |
> | HAC | 0.5989 |   0.5005 | 61.28 |
> | CoLin |  0.6212  | 0.5236 | 58.87|
> | $\epsilon$-greedy | 0.5977 | 0.4834 | 55.15|
> | ICM | 0.6057 | 0.5088 | 57.9 |
> | RND | 0.6012 | 0.5024 | 58.1 |
> | ECQL |  0.6313 |  0.5365 | 68.54 |
>
>
>
>
>
> **Q3: Why do online RL methods perform well in the offline learning setting, such as SAC, HAC in Table 2?**
>
> Thanks for this insightful question. We clarify that both SAC and HAC leverage an off-policy training paradigm, which is same as the proposed ECQL model. In particular, SAC maximizes the action predictive entropy in value estimates. HAC takes a hyper-action as an embedding to generate the actual action that selects the final recommendation list.  Both methods have been used in existing recommender systems and serve as baselines in multiple research works [3,4,5]. These works also simulate an off-policy training paradigm using the offline recommendation system datasets (e.g., MovieLens and Retailrocket).
>
> ---
> **References**
>
> [1] Pathak, Deepak, et al. "Curiosity-driven exploration by self-supervised prediction." International conference on machine learning. PMLR, 2017.
>
> [2] Burda, Yuri, et al. "Exploration by random network distillation." arXiv preprint arXiv:1810.12894 (2018).
>
> [3] Liu, Shuchang, et al. "Exploration and regularization of the latent action space in recommendation." Proceedings of the ACM Web Conference 2023. 2023.
>
> [4] Hong, Kimura, et al. "A Soft Actor-Critic Algorithm for Sequential Recommendation." In Database and Expert Systems Applications: 35th International Conference, DEXA 2024, Naples, Italy, August 26–28, 2024.
>
> [5] Silva, Á.L., Parra, et al. "On the Unexpected Effectiveness of Reinforcement Learning for Sequential Recommendation." Proceedings of the 41st International Conference on Machine Learning, 2024.

---

> > ### Author Response · Authors · 2024-11-29
> > **Follow-up from the authors and reminder of the reviewer-author discussion deadline [December 2nd]**
> >
> > Dear Reviewer mvAf,
> >
> > Thank you again for your thoughtful comments and questions! In the rebuttal, we have included comparisons with other RL exploration methods, including ICM and RND as suggested. Additionally, we clarified SAC and HAC simulate an off-policy training paradigm using the offline recommendation system datasets (e.g., MovieLens and Retailrocket), which is same as the proposed ECQL model. We hope you find our responses satisfactory and would be happy to address any further questions you may have.
> >
> > Sincerely,
> >
> > Authors

---

> > > ### Comment · Reviewer_mvAf · 2024-11-30
> > >
> > > Thanks for the response, I have raised the score to 8.

---

> > > > ### Author Response · Authors · 2024-11-30
> > > >
> > > > We sincerely thank the reviewer for taking the time to read our response and for your thoughtful feedback. We greatly appreciate your support of our work!

---

### Author Response · Authors · 2024-11-23
**General response**

We thank all reviewers for reviewing our paper and providing constructive comments and suggestions. Below, we summarize some of our major responses (MR):

---
**MR1: Motivation of proposed methodology.**

>Most existing methods treat recommendation as a static process, which prevents them from effectively accounting for users’ evolving preferences. Sequential recommendation methods address this limitation to some extent by leveraging previously interacted items to capture users’ dynamic behavior. However, prior RL-based recommender models largely rely on standard exploration strategies, such as $\epsilon$-greedy, which are less effective in scenarios with a large item space and sparse reward signals due to limited user interactions. As a result, these methods may struggle to learn an optimal policy that adequately captures users’ evolving preferences and achieves the maximum expected reward over the long term. The qualitative results presented by Figure 1 and Table 1 in the Introduction Section of the paper illustrate the limitations of existing approaches and further highlight the need for a systematic, uncertainty-aware exploration strategy.

>The proposed ECQL framework is designed to address the key limitations of existing RL based recommender systems. The novel evidential conservative Q-learning (ECQL) framework utilizes a balanced exploitation (with high predicted ratings) and exploration (with evidential uncertainty) strategy for effective recommendations. ECQL augments the reward based RL objective with **evidential uncertainty** to facilitate the exploration of unknown items. The evidential uncertainty formulation substantially improves exploration and robustness by acquiring diverse behaviors that are indicative of a user's long-term interest. Additionally, it encourages the model to explore items that do not significantly deviate from users' current interests given the sparse interactions. Such gradual exploration is guided by a conservative **evidential Q-value** that prevents recommending totally irrelevant items, causing user frustrations.

**MR2: Novelty of introducing evidence-based uncertainty.**

>Compared to existing uncertainty-based exploration strategies, we would like to clarify that the proposed Evidential Conservative Q-Learning (ECQL) framework is specifically designed to address a set of novel challenges unique to recommender systems, including:

1. Limited training data due to scarce user interactions,

2.  Sparse and noisy reward signals, and

3. Complex dynamics in user behavior, where long-term interests are intertwined with temporary preferences.

>Our framework introduces a novel approach by employing a more precise second-order uncertainty measure—vacuity—to reflect the lack of knowledge about candidate items in the item pool based on the current model analysis. This vacuity-based measure is leveraged to discover novel items that have the potential to align with users’ long-term interests. While effective exploration is crucial to accurately capturing users’ evolving preferences, ineffective exploration can lead to irrelevant recommendations that quickly erode user trust. To address these long-standing challenges, we propose a newly designed evidential off-policy reinforcement learning model. This model learns an effective and safe recommendation policy through a **non-trivial integration of evidence-based exploration and conservative learning**, providing a balanced approach to exploration and exploitation in recommender systems. Our comprehensive evaluation and through ablation study show that ECQL outperforms existing RL methods with uncertainty-based exploration strategies with a large margin, which further justifies the effectiveness of our novel exploration strategy.

**MR3: Discussion and comparison with other exploration methods.**

>As suggested by the reviewers, we have included a detailed discussion of other exploration methods, including ICM and RND. Additionally, we conducted further experiments to adapt these exploration strategies to the recommender system setting. The comparison results demonstrate that our proposed approach outperforms both ICM and RND in effectively capturing user preferences and guiding recommendations. These findings further validate the effectiveness of our method in addressing the challenges of sparse rewards and user interest shifts in recommender systems.

---
We have provided detailed responses to each individual comment from all reviewers and incorporated the suggested changes into the revised paper, with the updates highlighted in blue. We sincerely appreciate the insightful questions and feedback, which have helped us further refine and strengthen our work. We look forward to any additional feedback based on our responses. Thank you again for your valuable input!

Best,

Authors

---

### Author Response · Authors · 2024-11-29
**Summary of our responses and reminder of the reviewer-author discussion deadline [December 2nd]**

Dear Reviewers,

Thank you once again for taking the time to review our paper and provide thoughtful comments and suggestions. As the reviewer-author discussion deadline (December 2nd) approaches, we would like to summarize our key responses to your feedback:

- **Reviewer mvAf**:

  - Included comparisons with other RL exploration methods (ICM and RND) as suggested.
  - Clarified the settings for baseline models, including SAC and HAC.

- **Reviewer JFxC**:

  - Emphasized that ECQL includes only essential RL components, avoiding unnecessary complexity, while highlighting its key novelties.
  - Referenced our ablation study results to illustrate the contribution of each component.
  - Clarified the experimental settings.

- **Reviewer bGtM**:

  - Expanded on the motivation and novelty of our work.
  - Highlighted the versatility of our model as a plug-and-play component.

- **Reviewer dEvt**:

  - Provided a clear definition of "time step" in Equation 2.
  - Justified the fairness of our experimental setup.

We believe that these revisions have significantly improved our paper, and we deeply value your constructive feedback. Please let us know if there are any additional questions or concerns.

Sincerely,

Authors

---

### Meta-Review · Area_Chair_1pu3 · 2024-12-19

**Metareview:**

The submission proposes a novel Evidential Conservative Q-Learning (ECQL) framework aimed at addressing the exploration-exploitation tradeoff in reinforcement learning-based recommender systems. The ECQL integrates evidence-based uncertainty and conservative learning to optimize recommendations for long-term user interests while ensuring robustness and relevance. Two key components underpin this work: a sequential state encoder that models dynamic user behavior and a Conservative Evidential Actor-Critic (CEAC) module that balances exploration with user interest consistency. The extensive experimentation demonstrates ECQL's superiority across various benchmarks, achieving state-of-the-art results and effectively capturing long-term user preferences.

Strengths of the paper include its originality in applying evidential uncertainty to recommender systems, detailed experimental validation, and strong theoretical contributions, including the justification of the connection between evidential uncertainty and conservative policy updates. The reviewers generally appreciated the problem significance, methodological rigor, and comprehensive ablation studies that validated the contributions of individual components.

However, the paper has certain weaknesses. Some reviewers noted that introducing uncertainty into reinforcement learning is not novel, requiring the authors to clarify their distinct contribution to recommender systems specifically. Other issues raised include limited sequential behavioral patterns in the datasets, concerns about over-engineering, and questions about whether the model's components are all essential. These concerns were addressed in the rebuttal with additional experimental results, detailed clarifications, and revisions to the paper.

The most important reason for recommending acceptance is the paper’s solid contributions to addressing a critical challenge in RL-based recommender systems—effectively capturing user preferences over the long term. The authors successfully defended the novelty and utility of their approach through thoughtful responses, additional experiments, and improved clarity in their rebuttal. While some concerns about novelty and generalizability were raised, the majority of the reviewers agreed that the work represents a meaningful advancement in this domain.

**Additional Comments On Reviewer Discussion:**

During the discussion, most reviewers acknowledged the authors' efforts in addressing their concerns. Reviewer mvAf initially raised issues about the lack of comparisons with exploration methods such as ICM and RND but raised the score after the rebuttal clarified these points and added results demonstrating ECQL's advantages. Reviewer JFxC expressed concerns about the model's complexity and clarity, questioning the necessity of all components. While the authors provided ablation studies and defended the model’s design, Reviewer JFxC maintained a more critical stance. Reviewers bGtM and dEvt were largely supportive after the rebuttal clarified issues such as the definition of "time step" and the motivation for the proposed exploration strategy.

The discussion highlighted varying reviewer perspectives on the paper's novelty and complexity, but most concerns were addressed satisfactorily. The final consensus, reflected in the reviewers' scores and comments, supports the paper's acceptance, with most reviewers appreciating its contributions and experimental rigor. As the Area Chair, I weigh the thoughtful responses and majority agreement heavily in favor of acceptance.

---

### Decision · Program_Chairs · 2025-01-22

Accept (Poster)